# Elevation dependent spatial interpolation of hourly rainfall for accurate flood inundation modelling

Chi Nguyen[1], Jai Vaze[1], Cherry May R. Mateo[1], Michael F. Hutchinson[2], and Jin Teng[1]

[1]CSIRO Environment, Australian Capital Territory, Australia
[2]Australian National University, Australian Capital Territory, Australia

**Correspondence:** Chi Nguyen (chi.nguyen@csiro.au)

**Abstract.** High-quality rainfall data are crucial for various climatological and hydrological applications, especially in detailed modelling at fine temporal and spatial resolutions. However, obtaining precipitation data with fine spatiotemporal resolution is often challenging due to the limited availability of sub-daily point measurements and the sparse distribution of rainfall stations in many regions. This paper presents and demonstrates a method to generate the Commonwealth Scientific and Industrial Research Organisation Hourly Rainfall (CHRain) dataset, which provides hourly and 1 km gridded rainfall surfaces for hydrological/hydrodynamic modelling. The method applies thin-plate spline interpolation to generate rainfall surfaces using hourly input time series obtained from hourly rainfall stations, and from daily data disaggregated into hourly intervals based on patterns observed in nearby hourly rainfall stations, and also guided by continuous radar images. The method is used to represent rainfall patterns and amounts from 2007 to 2022 in the Richmond River catchment in New South Wales, Australia. Our analysis shows that the performance of the spline interpolation improves with the inclusion of the elevation data. Larger rainfalls responded more sensitively to changes in topography, with an optimum supporting DEM horizontal resolution of around 5 km, in agreement with previous studies. Performance was also significantly enhanced by using a stable spatial occurrence analysis to reliably remove false zeroes from the data. About 0.26% of the data were found to be false zeroes. During the 2017 event, CHRain achieved a correlation coefficient of 0.949 against hourly gauges, showing that the dataset can adequately reproduce the patterns of hourly rainfall measurements. The spatial and temporal analyses indicate that the CHRain dataset outperforms other gridded datasets currently available in Australia in representing the sub-grid distribution, the daily and hourly variation of rainfall across the study area, and the high rainfall values. These are all essential for capturing the spatiotemporal characteristics of flood inundation in the study area, which is frequented by disastrous flood events.

## 1 Introduction

High resolution temporal and spatial representations of precipitation data are required in many hydrological applications, such as modelling flood inundation (Jhong et al., 2017; Pappenberger et al., 2005), analysing catchment responses in rainfall-

runoff models (Xu et al., 2022; Acharya et al., 2019), and forecasting extreme events and natural hazards (Ficchi et al., 2016; Mukherjee et al., 2018). Sub-daily and even sub-hourly precipitation data are required to accurately represent the variability of rainfall especially during extreme flood events or when a catchment receives excessive and intense amount of rainfall within a few minutes to several hours (Davis, 2001; Ficchi et al., 2016; Westra et al., 2012). Several studies (Ficchi et al., 2016; Acharya et al., 2022; Brighenti et al., 2019) indicated that improving the quality of rainfall data temporally can enhance the performance of rainfall-runoff models in simulating flood peaks, flood frequency, and the timing of the peaks. Peleg et al. (2013) analysed the subpixel rain distribution by comparing the data from radar with point measurements at high density gauges. The results showed that a density of 3 rain gauges per radar pixel (4 km $\times$ 4 km) will allow an adequate presentation of radar rainfall. Peleg et al. (2017) indicated a valuable contribution of 26% of spatial distribution rainfall on the total variability of modelled urban drainage network. However, high spatial and temporal resolution precipitation data are not always available for those applications.

There are significant variations in the rainfall patterns in Australia at both regional and seasonal scales (Taschetto and England, 2009). The rainfall patterns can be observed from rainfall time series measured at stations and gridded data with various resolutions. There are more rainfall stations that record at daily intervals than those that record at hourly or sub-hourly intervals. Observations from daily stations are also available for longer periods than the hourly stations. There are 4765 active daily rainfall stations with data from the 1960s in Australia. There are 759 sub-daily rainfall stations and only 442 stations having records more than 20 years long (Morbidelli et al., 2020; Westra et al., 2012). Most rainfall stations are located in highly populated regions such as the southwest, east-coastal, and south-coastal areas (Morbidelli et al., 2020). The coarse distribution of rainfall stations in some regions and the short records of available data limit the ability to generate sub-daily rainfall data at a high spatial resolution for the whole of Australia.

Some efforts have been invested in disaggregating daily rainfall data to sub-daily (Acharya et al., 2022; Schreider and Jakeman, 2001; Breinl and Di Baldassarre, 2019). Acharya et al. (2022) disaggregated daily rainfall data from the Australian Gridded Climate Data (AGCD) version 1 (previously known as Australian Water Availability Project (AWAP) (Jones et al., 2009)) to hourly using the patterns from a coarser spatial resolution dataset of the Bureau of Meteorology Atmospheric high-resolution Regional Reanalysis for Australia (BARRA) (Su et al., 2019). Westra et al. (2012); Breinl and Di Baldassarre (2019) applied the method of fragments, which finds the relationship between hourly and daily data of the currently available records and applies a moving window to disaggregate the daily data where the hourly data are not available. A comparison by Pui et al. (2012) showed that the method of fragments resulted in a better performance in keeping intensity-frequency relationships at the hourly scale and disaggregating extreme values than other parameterized methods, such as the random multiplicative cascades and the randomized Bartlett–Lewis model. These disaggregation methods open options to produce sub-daily time series at a higher temporal resolution.

Although daily rainfall measurements are reliable and available for a reasonably long period in Australia (although at limited spatial locations), many hydrological applications require gridded rainfall data to present the rainfall variation over land surfaces (e.g., detailed climate inputs for hydrological and hydrodynamic models). Several techniques have been applied to generate spatial rainfall data in Australia. There are three common types of gridded rainfall data based on point measurements,

satellite data, and model reanalyses (Chua et al., 2022). The thin-plate spline interpolation method has been widely applied
to generate daily, monthly to mean annual rainfall surfaces (Hutchinson, 1995; Johnson et al., 2016; Hutchinson et al., 2009).

Thin-plate spline interpolation allows the inclusion of topography patterns, which has been shown to have a significant impact
on the spatial distribution and quantity of rainfall (Johnson et al., 2016). This method was applied to generate the ANUCli-
mate dataset, which is the daily and 0.01°resolution (approximately 1 km) climate gridded data, including daily rainfall from
1900 for the whole of Australia (Hutchinson et al., 2021). Jeffrey et al. (2001) interpolated ground measurement data using
ordinary kriging to generate the climate surfaces of Scientific Information For Land Owners (SILO) including daily rainfall

at 0.05°grid. The AWAP dataset also provides daily and monthly spatial rainfall at a resolution of 0.05°(Jones et al., 2009).
The AWAP dataset are generated using an anomaly-based method, including the application of Barnes successive correction
method (Jones and Trewin, 2000) to generate weighted-anomalies layers at daily time steps, and thin spline interpolation to
provide the relationship between point measurements and locations (longitude, latitude and elevation) (Jones et al., 2009). The
AWAP data was enhanced to produce the AGCD dataset, using statistical interpolation and satellite rainfall data (Chua et al.,

2022). However, Chappell et al. (2013) indicated no clear benefit of blending satellite data with point measurements compared
with ordinary point kriging in estimating near real-time rainfall in Australia. The satellite data only appeared to improve rainfall
estimation where the distribution of rainfall stations is sparse (e.g., less than 4 gauges per 10,000 km$^2$)(Chappell et al., 2013).
Instead of using observation such as point measurements or satellite data, the reanalysed rainfall data are usually generated
from models solving deep-atmosphere global non-hydrostatic equations (Wood et al., 2014). BARRA is the first gridded dataset

providing hourly rainfall data for the Australasian region at approximately 12 km resolution, with a downscale sub-product of
1.5 km resolution in 4 areas. The evaluation by Acharya et al. (2019) showed that reanalysed rainfall data (i.e., from BARRA)
had poorer performance compared to interpolated rainfall data (i.e., from AWAP) in terms of representing the point measure-
ments. Lewis et al. (2018) applied a nearest neighbour interpolation scheme to disaggregate 1 km gridded estimates of daily
and monthly areal rainfall for the United Kingdom (CEH-GEAR) to produce an hourly dataset. However, the method is not

applicable in Australia for several reasons. The distribution of hourly rainfall gauges in Australia is much coarser, especially
in the central and northern parts of Australia, compared with the distribution in the United Kingdom. The record of hourly
measurements is shorter than the daily data and only available from 2007; therefore, a method to disaggregate daily rainfalls to
hourly when there is no or very little hourly observations is needed before we can disaggregate gridded data for those periods.
Despite all the efforts, there are still gaps in generating high resolution temporal and spatial rainfall data, which are relevant to

hydrological purposes, especially for detailed flood modelling using fully distributed hydrodynamic models.

An accurate high spatial and temporal resolution rainfall is a critical input for accurately representing flood volumes and
times of flood peaks. This paper presents a method to generate the Commonwealth Scientific and Industrial Research Organi-
zation Hourly Rainfall (CHRain) dataset, which consists of high temporal (hourly) and spatial resolution (1 km grids) rainfall
surfaces that capture the sub-daily instantaneous variation of rainfall patterns, necessary for modelling heavy rainfall events.

The method uses hourly point rainfall measurements and thin-plate spline interpolation to generate hourly rainfall surfaces
at 1 km resolution. In the areas with sparse distribution of hourly rainfall stations, daily measurements are disaggregated to
hourly data from 9:00 am the previous day to 8:00 am the current day, using patterns from nearby hourly rainfall stations,

to match with the daily data provided by the Australian Bureau of Meteorology (BoM). We applied the proposed method to produce hourly rainfall surfaces for the Richmond River catchment ($\approx 7025$ km$^2$) in New South Wales, Australia. The new rainfall surfaces are evaluated using point measurements and other common gridded datasets currently available in Australia. The method proposed in this study opens an opportunity to produce high resolution spatiotemporal rainfall surfaces for other regions where detailed modelling is to be undertaken.

## 2 Data and methods

The study area is the Richmond River catchment, located in the northern rivers region of New South Wales, Australia, near the border between New South Wales and Queensland (Fig.1). The catchment area is approximately 7025 km$^2$. The north and west sides of the catchment are mostly forested, while the central to the south-east areas are agricultural land (NSW Department of Planning & Environment, 2024). The topography of the catchment changes significantly across the landscape. The elevation ranges between 0 m and 934.6 m across the catchment. Most of the northern and western mountainous areas and the areas upstream of Lismore are very steep, while the southern and the coastal areas around Casino are very flat. The Richmond River catchment is an important habitat for endangered fauna and flora. The national parks and reserves, e.g., the Border Ranges, are protected under the Australia World Heritage (NSW Department of Planning & Environment, 2024).

The annual rainfall in the catchment can exceed 1800 mm per year, with particularly high rainfall intensities observed in the north-east and coastal areas (Lerat et al., 2022). Due to the combination of the topographic and climate conditions, the Richmond River catchment is prone to extreme and devastating floods. There were 17 major flood events from 1945 to 2022, with a maximum daily rainfall of more than 60 mmd$^{-1}$ (Lerat et al., 2022). The severe floods in 2017 (1 in 21 Annual Exceedance Probability (AEP)) and 2022 (the largest observed flood event in the catchment on record) overtopped the levee at Lismore, causing loss of lives and serious damages to businesses and properties. Having a more precise representation of the rainfall data in the Richmond River catchment is essential for reliable flood modelling and mitigation in the region. The analysis was done for an area (30,389 km$^2$) as shown in Fig. 1, which is larger than the Richmond River catchment area, to adequately support the hourly rainfall interpolation along the catchment boundaries.

### 2.1 Thin-plate spline interpolation model

The thin-plate smoothing splines method, as described by Wahba (1990), fits a "smooth" function to a set of noisy data across a multidimensional space. Hutchinson (1995) applied the method to generate surfaces of climate variables such as temperature, rainfall, and evaporation, while considering the impacts of topographic conditions. The model for thin plate spline interpolation is:

$$z_i = f(x_i) + b^T y_i + e_i \text{ for } i = 1,...,N; \tag{1}$$

where $x_i$ is a d-dimensional vector of spline independent variables; $y_i$ is a p-dimensional vector of independent covariates; $z_i$ is the value of a data point at location $x_i$; $f$ is an unknown smooth function of $x_i$; $b$ is an unknown p-dimensional vector of

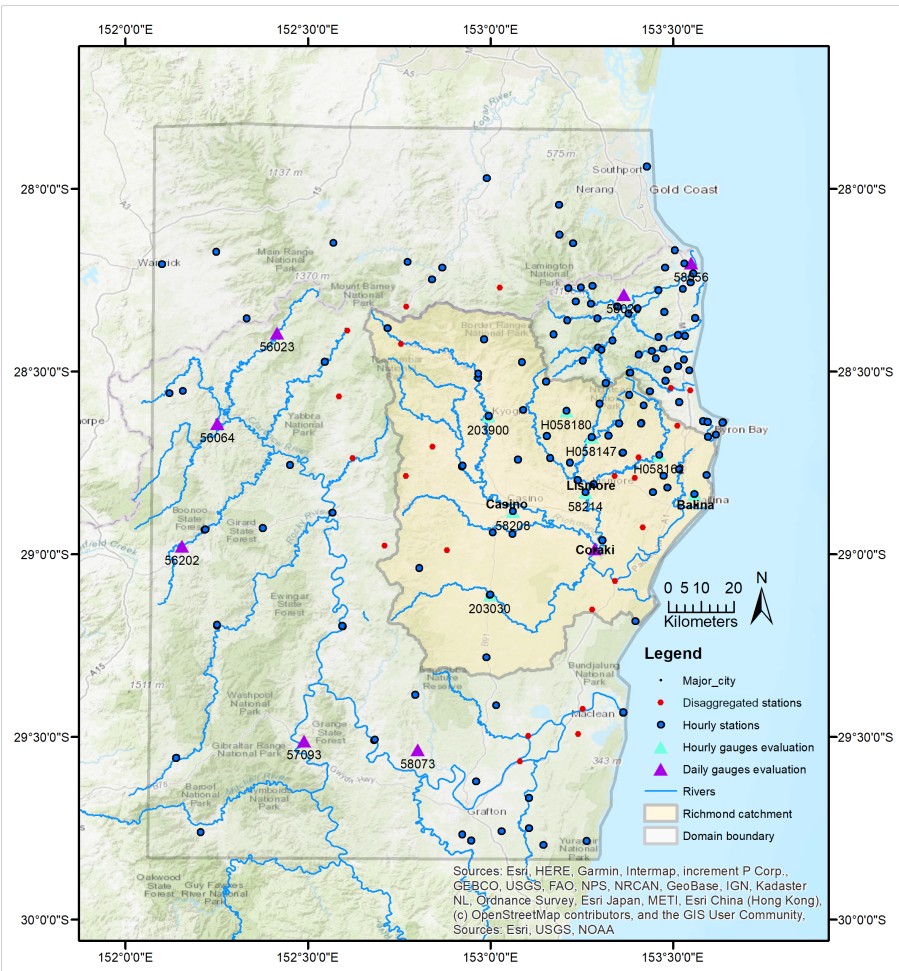

**Figure 1.** Locations of the study area at Richmond catchment.

coefficients; $e_i$ is the independent zero mean error with variance $\sigma^2$ that is constant across all data points; and N is the total
number of observed data. The smooth function $f$ and coefficient $b$ are found by minimising the function below:

$$\sum_{i=1}^{N}[z_i - f(x_i) - b^T y_i]^2 + \rho J_m(f) \tag{2}$$

where $\rho J_m(f)$ is a measure of the complexity of $f$, which is an integral of $m^{\text{th}}$ order partial derivatives of $f$, and $\rho$ is a positive
smoothing parameter. The smoothing parameter is normally determined by minimising the generalised cross validation, a
measure of the mean square predictive error of the fitted spline function.

In this analysis, we employed the software ANUSPLIN Version 4.4 to generate hourly rainfall trivariate spline functions of longitude, latitude and appropriately scaled elevation. The elevations were obtained from DEMs with a range of underpinning horizontal resolutions. The detailed description of the setup and input files is available in Hutchinson and Xu (2004).

## 2.2 Rainfall data

In our analysis, we used the daily and hourly point rainfall measurements to interpolate the rainfall surfaces and to generate the gridded datasets. In areas where the distribution of hourly gauges is coarse, the rainfall data at nearby daily gauges were disaggregated to hourly (by using the hourly patterns from the neighbouring hourly stations and radar images to determine the rain front movement) for the spline interpolation. Gridded rainfall datasets, including the radar, BARRA data for the eastern New South Wales (BARRA-SY), ANUClimate, and AGCD datasets, were used in the evaluation and comparison with our results from CHRain (Table 1).

**Table 1.** Gridded data descriptions

| Dataset | Description | Method | Domain | Resolution | Reference |
|---------|-------------|--------|--------|------------|-----------|
| BARRA-SY | Bureau of Meteorology Atmospheric high-resolution Regional Reanalysis for the Eastern New South Wales, 1990 - 2019 | Local reanalysis | ([-28°, -38°], [147°, 155°]) | hourly, 1.5 km | Su et al. (2019) |
| Radar | Radar-Derived Rainfall Accumulations, 2013 - present | Radar blended | 128 km radius centred around the radar location at Grafton (-29.62°, 152.97°) | hourly, 1 km | Bureau of Meteorology (2023) |
| ANUClimate | Australian National University Climate, 1900 - present | Gauge interpolation | Australia land area | daily, 1 km | Hutchinson et al. (2021) |
| AGCD/AWAP | Australian Gridded Climate Data / Australian Water Availability Project, 1900 - present | Gauge interpolation | Australia land area | daily, 5 km | Jones et al. (2009) |

The daily and hourly data at rainfall gauges were sourced from the BoM and the Water New South Wales Corporation (WaterNSW). The rainfall data during the flood events in 2022 at Rocky Creek Dam (RCD) and Emigrant Creek Dam (ECD) were provided by the Rous County Council. These two stations are critical as both these stations are located in the higher rainfall areas where there are limited gauges. There are 330 daily stations with records from 2007 to 2022. However, only 253 stations are active in 2022. There are 143 hourly rainfall stations. Most of the hourly records start from 30/01/2007. A detailed

quality control was undertaken for all the rainfall data before being used in the ANUSPLIN program to construct the CHRain hourly rainfall surfaces at 1 km resolution, from 30/01/2007 to 31/12/2022.

The radar data were provided by the BoM, showing the rain front movement and rainfall intensity over the catchment area (image every 5 minutes). The radar intensity data were used to generate Radar-derived rainfall accumulation, showing the amount of rainfall accumulating in 1 hour (Bureau of Meteorology, 2023). We acknowledge that the radar rainfall shows the

rainfall in the atmosphere instead of the rainfall reaching the ground. There are errors in the radar-derived rainfall data, showing unreasonable high rainfall values in some areas (Bureau of Meteorology, 2023). The radar data were employed to observe and understand the movement and distribution of rainfall front in the study area. The hourly rainfall accumulating from radar data were not used in our analysis. The radar images in the Richmond River catchment were available from 2/12/2013.

The hourly 1.5 km resolution BARRA-SY dataset was compared with our hourly CHRain product. The BARRA-SY

dataset is available from 01/01/1990 to 28/02/2019 and covers a domain with the latitude range [-28°, -38°] and the longitude range [147°, 155°]. The ANUClimate version 2 dataset (Hutchinson et al., 2021) provides gridded daily rainfall data at 0.01°resolution (approximately 1 km) from 01/01/1900. These grids have been generated using the thin-plate spline method to interpolate daily point measurements, considering the impacts of topography (Hutchinson, 1995; Johnson et al., 2016). The AGCD dataset contains daily 0.05°resolution (approximately 5 km) rainfall surfaces from 01/01/1900. The AGCD dataset

covers the whole of Australia and is regularly updated with real-time data. The BARRA-SY, ANUClimate, and AGCG data are available from the National Computational Infrastructure Data Catalogue (https://geonetwork.nci.org.au/geonetwork/srv/eng/catalog.search#/home).

## 2.3   Quality control for the hourly rainfall data

A commonly used quality control method described in (Westra et al., 2014) was applied to the hourly and daily point rainfall

measurements. The first step checks the range of values and the changes overtime. We manually plotted rainfall time series and compared them to all neighboring stations. Thresholds of 300 mmh$^{-1}$ and 1500 mmd$^{-1}$ were used to remove unreasonably high hourly and daily rainfall data. The suspicious data were removed, including negative, unreasonable high values, linear interpolated values, and the values that were significantly higher or lower compared with those at nearby stations (within 5 km) and are also inconsistent with the radar data. Some unusually high values of hourly rainfall, mostly occurring at midnight,

were detected. If an hourly rainfall value exceeded the sum of the previous 23 hours by more than 30 mmh$^{-1}$, it was removed. Additionally, if there were two or more stations within 2 km of each other, they were compared, and only the more reliable one was retained (based on the quality code). This step is required to avoid instability in thin-plate spline interpolation, which occurs

when the close data points have very different rainfall values. The data from nearby stations were compared and combined if the rainfall records overlapped. The station with a longer record was retained to be used in the ANUSPLIN package.

Close inspection of initial analyses of the hourly rainfall data indicated that there were significant numbers of false zeroes in the data leading to underestimation of rainfall during periods of high rainfall. This is a common problem with rainfall data, particularly when they are recorded automatically. These values are hard to detect by applying simple thresholds. As noted by Hutchinson et al. (2009), rainfall occurrence is more spatially coherent than rainfall amounts. An initial trivariate spline analysis of the hourly occurrence data was therefore conducted to detect and automatically remove false zeroes.

Positive rainfalls were set to an occurrence value of 1 and zero rainfalls were set to an occurrence value of 0. The spline analysis used the same underpinning DEM resolution and elevation scaling as optimised for the rainfall amount analysis. Zero hourly rainfall values were deemed to be false, and removed from the data set, when the interpolated occurrence value exceeded 0.5. The limited spatial coverage of the data set led to instabilities when the data values were almost all positive or almost all zero. This was overcome by setting a constant error standard deviation of 0.25, consistent with the automatically derived error

standard deviations when there were significant numbers of zeroes and ones. This ensured that sufficient smoothing was applied to the data to interpolate spatially stable occurrence patterns with a robust dependence on the data values. A total of 42,193 false zeroes were removed from a total number of 15,737,817 data values, amounting to 0.26% of the data.

The reliability of the occurrence based corrections was assessed by comparing the analyses with hourly radar rainfall data over eight successive days during the two flood events in 2022 (with the peak around 28/02/2022 and 29/03/2022). Summary

statistics are presented in Table 2. As noted above, the radar rainfall is not always reliable. The percent occurrence agreement of the raw hourly rainfall data with the radar data ranged between 56% and 72% for six of the eight days, while there were strong occurrence agreements of 92% and 81% on the two high rainfall days on 27/02/2022 and 28/03/2022. This indicates there were major deficiencies in the radar data, except on the heavy rainfall days when significant rainfall was widespread over the data network. Comparing the occurrence corrections with the radar occurrence data showed strong agreement with the original

data occurrence agreements, ranging from 43% to 92% on six of the days and 98% and 83% on 27/02/2022 and 28/03/2022. If the corrections were all correct, comparison with the radar data could be expected to assess them as having an accuracy similar to the initial overall agreements between the rainfall data and the radar data. The strong agreement between column 3 and column 5 in Table 2 is consistent with the corrections being in fact highly reliable, with a true accuracy up to around 98% on all eight days. The true reliability maybe somewhat lower on days with less widespread rainfall and less spatially coherent

rainfall occurrence patterns. The occurrence corrections were sufficient to improve the overall occurrence agreement with the radar data on the fourth day in the first event and on all three days in the second event. The overall agreement was unchanged for the other days. A range of occurrence thresholds was tested by assessing the overall occurrence agreement of the corrected data with respect to the occurrence threshold. A range of thresholds from 0.4 to 0.8 gave similar results to the chosen value of 0.5. Further refinements are limited by the overall unreliability of the radar rainfall data.

Close inspection of the analyses and a detailed comparison with radar data over high rainfall periods indicated that the false zero detections are reliable. The occurrence analysis is also illustrated by tabulating the occurrence corrections for a high rainfall day in Appendix B.

**Table 2.** Summary statistics of occurrence corrections with respect to hourly radar data over five successive days, during the two flood events in 2022. Occurrence agreement is calculated as the percentage of the number of agreements in occurrence (both zero or both non-zero) divided by the total number of hourly data values.

| Date | Average rainfall [$mmh^{-1}$] | Percent occurrence agreement of raw data with radar data | Number of estimated false zeros | Percent agreement of corrections with radar data | Percent occurrence agreement of corrected data with radar data |
|---|---|---|---|---|---|
| 24/02/2022 | 3.3 | 57 | 75 | 57 | 57 |
| 25/02/2022 | 0.8 | 57 | 36 | 53 | 57 |
| 26/02/2022 | 2.3 | 72 | 54 | 87 | 72 |
| 27/02/2022 | 8.6 | 92 | 102 | 98 | 96 |
| 28/02/2022 | 10.5 | 56 | 85 | 46 | 56 |
| 28/03/2022 | 4.2 | 81 | 90 | 83 | 83 |
| 29/03/2022 | 4.2 | 65 | 94 | 92 | 67 |
| 30/03/2022 | 3.0 | 57 | 83 | 43 | 58 |

## 2.4 Disaggregation of daily rainfall data to hourly rainfall data

The distribution of hourly stations in the Richmond River catchment is sparse in some areas, especially at the west boundary of the catchment. We chose 23 daily rainfall stations (shown as red dots in Fig. 1) to disaggregate the rainfall data from daily to hourly, using the patterns from the nearest hourly stations. We also used the observed movement of rainfall from the radar data to select suitable nearby hourly gauges to disaggregate data from daily to hourly.

Some criteria were set up to disaggregate daily data into hourly:

1. The daily rainfall data were disaggregated using the hourly distribution pattern from the nearest hourly station. The summed 24-hour hourly data from 9:00 am the previous day to 8:00 am of the current day was scaled to match the daily recorded total for that day.

2. If a daily record at a certain time step was missing (no data), the associated 24-hour data were set as missing values in the disaggregated dataset.

3. If a daily record at a certain time step was positive but the hourly data on the same day at the nearby station were missing or 0, the daily rainfall value was distributed equally over 24 hours.

After cleaning, disaggregating, and completing a detailed quality control of the data, there were 139 hourly stations (including 23 disaggregated stations) for generating hourly rainfall surfaces (shown in Fig. 1).

## 2.5 Calibration of the DEM smoothing scale and the elevation transformation parameter

The 5 m resampled to 1 km averaged LiDAR Digital Elevation Model (DEM) from Geosciences Australia was used to define
the boundary of the rainfall surfaces in the ANUSPLIN package (https://ecat.ga.gov.au/geonetwork/srv/eng/catalog.search#/metadata/89644). A set of 1 km resolution smoothed DEMs was prepared by calculating the focal mean with distances from 2 to 10 km to investigate the impacts of topographic scale on the rainfall surfaces using ArcGIS program. The focal mean at each 1 km pixel is calculated as the mean of a square window with a specified distance around that pixel.

In the ANUSPLIN program, the independent variable transformation for the DEM is $h/a$, where $h$ [m] is the elevation value and $a$ is the transformation parameter. The usual recommended $a$ value for interpolating monthly and daily data is 1000 (Hutchinson, 1995; Hutchinson et al., 2009). This corresponds to a 100-fold exaggeration of the impact of elevation on precipitation patterns compared to the impact of horizontal position. In this study for hourly splines, $a$ was calibrated in the range from 1000 to 10,000, corresponding to vertical exaggerations ranging from 100-fold to 10-fold. We also tested the performance of the interpolation model using bivariate (without the elevation variable) and trivariate (with the elevation variable) analyses.

The days of hourly rainfall data were categorised into two groups to analyse the impact of topography on spatial rainfall patterns. Days with average hourly rainfall between 0 and 1 mmh$^{-1}$ were considered as light rain days, and days with average hourly rainfall exceeding 1 mmh$^{-1}$ were considered medium to high rainfall days. There were 3379 light rainfall days and 111 medium to high rainfall days. There were 246 days with zero rainfall across the whole data network. These days were omitted from the calibration. The focal mean distance and the elevation scaling parameter $a$ were jointly optimised to minimise the average of the generalised cross validation of the fitted splines over all medium to high rainfall days.

The performances of the different spline models were compared using the Mean Absolute Predictive Error (MAPE) and the Mean Absolute Residual (MAR) provided by the spline interpolation model. The MAPE is calculated from the individual cross validation residuals as afforded by the "leaving out one lemma" described in Wahba (1990).

## 2.6 Generate hourly splines using ANUSPLIN

The hourly rainfall splines were generated using ANUSPLIN version 4.4 (Hutchinson and Xu, 2004). There are four main steps to generate daily and hourly splines, including preparing the input data (.dat) files, preparing the command (.cmt) files, running the **spline** program to generate interpolating parameters, and running the **lapgrd** program to generate rainfall surfaces. For the hourly rainfall surfaces, we ran the ANUSPLIN program daily (24 splines per day) from 30/01/2007 to 31/12/2022. The details of the setup are:

1. The independent variables include the longitudes, latitudes, and DEM values of the hourly stations. The dependent variables are the measured rainfall values at the hourly stations.

2. For the **spline** commands, the numbers of knots were set as 90% of the total number of stations, as read from the input data files. The dependent variable transformation was set as the square root of the data surface to comply with

the positive skew of the rainfall values, often including many zeroes, and to ensure that the fitted values are always non-negative Hutchinson et al. (2009).

3. The optimised parameters from the **spline** program and the 1 km smoothed DEM were input into the **lapgrd** program to generate the rainfall grids.

## 2.7 Temporal and spatial analyses

### 2.7.1 Temporal analysis

We calculated the statistics for the hourly rainfall record during the simulation period from 2007 to 2022, including the mean, maximum, standard deviation, and the ratios of the maximum values at different accumulated time intervals (i.e., 3, 6, 12, and 24 hours) to the maximum values in the hourly time series ($P_k[\%]$):

$$P_k = \frac{\max\limits_{i=1}^{N}\left(\frac{1}{k}\sum\limits_{i-\frac{k-1}{2}}^{i+\frac{k-1}{2}} x_i\right)}{\max\limits_{i=1}^{N}(x_i)} \times 100, \tag{3}$$

where $x_i$ is the hourly rainfall value at time step $i$, $k$ is the rolling sum time interval (3, 6, 12, and 24 hours), and N is the total
number of observed data.

Since hourly rainfall data usually contains numerous zero values, the evaluation metrics calculated for a long period are biased toward underestimation of extreme values (Gires et al., 2012). Therefore, the flood event in 2017 (1 in 21 AEP) and in 2022 (the biggest flood event observed in the catchment) were selected for further evaluation. The flood event in 2017 started from 01/03/2017 to 05/04/2017, with the peak rainfall period occurring on 30-31/03/2017. The flood event in 2022 occurred
from 25/01/2022 to 05/05/2022, including two peak events on 28/02 - 01/03/2022 and 29-30/03/2022. The thresholds of 0.1 mmh$^{-1}$ and 1 mmd$^{-1}$ were used to eliminate the numerical noise in the interpolated splines and to classify dry and wet pixels.

In the temporal evaluation, we compared the time series extracted from gridded rainfall data, including CHRain, BARRA-SY, radar, ANUClimate, and AGCD datasets to the point measurements. Because all of the hourly gauges were included in the generation of the CHRain dataset, we evaluated the CHRain with the daily measurements at 169 gauges, that were not used in
the interpolation. We selected 8 hourly stations to undertake further analysis, shown as blue triangles in Fig. 1. The 8 gauges are located in the important cities and towns within the Richmond Rivers catchment, including Lismore, Casino, Ballina, Kyogle, Channon, and Nimbin. These areas were affected significantly during the flood events in 2017 and 2022. The ANUClimate and AGCD daily values were disaggregated evenly from 9:00 am the previous day to 8:00 am the current day to generate the hourly time series. A similar comparison was conducted for the daily time series, extracted from 8 daily stations (shown as purple
triangles in Fig. 1). These daily stations were not used in generating the CHRain splines. The hourly CHRain, BARRA-SY, and radar data were aggregated from 9:00 am the previous day to 8:00 am the current day to produce the daily datasets to compare with ANUClimate and AGCD data.

The Bias, Mean Absolute Error (MAE), correlation coefficient ($r$), Nash–Sutcliffe Efficiency (NSE) metrics were calculated in the evaluation (Appendix A). Positive and negative bias values show overestimation and underestimation, respectively. The

MAE shows the absolute errors of the predicted values compared to the measurement data. The range of the NSE is from $-\infty$ to 1, where 1 is the optimal value.

### 2.7.2 Spatial analysis

In the spatial analyses, we compared the hourly CHRain with the ANUClimate and AGCD datasets. The hourly CHRain data were summed to generate 24-hour total surfaces, from 9:00 am the previous day to 8:00 am the current day.

The daily rainfall data were classified as heavy and extremely heavy if the recorded values were higher than 95[th] and 99[th] percentiles of the daily measurement data from 2007 to 2022, as suggested by Bureau of Meteorology (2024). In the Richmond River catchment, rainfall values from 21 mmd[-1] to 58 mmd[-1] are considered heavy rain, and rainfall values higher than 58 mmd[-1] are classified as extremely heavy rainfall.

The Bias, Hit Rate, and the Critical Success Index (CSI) (Ebert, 2008) were used to compare the 24-hour total CHRain with

the ANUClimate. The optimal value for the Hit Rate and CSI is 1, showing a perfect match between the two datasets. The Bias value describes the difference between the generated grid and the observed data. The Hit Rate shows the proportion of wet pixels in the generated dataset that are correctly predicted. The CSI considers both the underestimation and overestimation of the generated dataset.

## 3 Results

### 3.1 Rainfall statistics

The statistics of the hourly rainfall time series from 30/01/2007 to 31/05/2022 are shown in Table 3. The maximum values during the 2017 flood event in the Richmond River catchment vary from 57.2 to 93.4 mmh$^{-1}$ in 8 hourly validated gauges. By averaging the hourly data from 3 to 24 hours, the dynamic extreme variation of the hourly rainfall is diminished. The averaged maximum rainfall values reduce from 62.6% to 26.2% if the averaging time interval increases from 3 hours to 24 hours (Table

3). Especially at station 203030, the peak of 24-hour averaged data can only capture 14.8% of the hourly peak value. Many hydrological applications, such as detailed hydrodynamic models, require hourly or even sub-hourly data to generate flows and water movement correctly, while the input rainfall is only usually available at a daily time step. If the daily rainfall totals are available and provided as input, the model disaggregates it evenly and distributes it over the day. This process leads to the underestimation of the hourly flood peaks. During flood events, intensive rainfall periods only occur over a few hours. Hence,

generating hourly rainfall data is essential to preserve the sub-daily variations in rainfall intensity and dynamic patterns of rainfall observations (Westra et al., 2014).

**Table 3.** Statistics for the observed hourly rainfall from 2007 to 2022.

| Station ID | Mean [mmh$^{-1}$] | Max [mmh$^{-1}$] | Std [mmh$^{-1}$] | $P_3$ [%] | $P_6$ [%] | $P_{12}$ [%] | $P_{24}$ [%] |
|---|---|---|---|---|---|---|---|
| 58214 | 1.6 | 57.2 | 3.3 | 72.0 | 50.0 | 35.3 | 32.3 |
| 203900 | 1.5 | 78.0 | 2.8 | 67.0 | 54.0 | 44.7 | 33.8 |
| 58198 | 2.1 | 93.4 | 4.0 | 46.4 | 28.2 | 21.5 | 14.7 |
| H058147 | 1.8 | 83.6 | 3.6 | 67.4 | 44.5 | 39.6 | 35.0 |
| 58208 | 1.7 | 61.6 | 3.3 | 94.0 | 65.9 | 40.8 | 40.8 |
| H058180 | 1.6 | 58.6 | 3.1 | 50.9 | 32.8 | 28.9 | 15.8 |
| H058162 | 1.8 | 70.9 | 3.4 | 46.5 | 42.1 | 30.1 | 22.6 |
| 203030 | 1.8 | 84.4 | 3.6 | 56.2 | 39.3 | 22.0 | 14.8 |
| **Average** | **1.7** | **73.5** | **3.4** | **62.6** | **44.6** | **32.9** | **26.2** |

### 3.2 Impacts of topography on the spatial interpolation of hourly rainfall splines

Table 4 and Table 5 show the Square RooT of the average Generalised Cross Validation (RTGCV) of the trivariate spline model for light rainfall days and medium to high rainfall days as a function of DEM focal distance and elevation scaling, as derived in the initial analyses with no removal of false zeroes. The light rainfall days indicate a very broad dependence on the topographic parameters with an optimum DEM focal distance around 10 km or possibly larger. On the other hand, the medium to high rainfall days indicate an optimum DEM focal distance of around 5 km and an optimum elevation scaling of around 4000. This suggests that topography plays an important role in interpolating larger rainfalls while the response of smaller rainfalls to topography is fairly flat. The daily average 1 mmh$^{-1}$ threshold appears to be an effective discriminator of light and medium to high rainfall days. Setting a lower threshold gave rise to multiple local minima in the RTGCV patterns for days with average hourly rainfall greater than 0.5 mmh$^{-1}$. These tables were recalculated after false zeroes were removed by the spline occurrence analysis described above, with DEM focal distance set to 5 km and elevation scaling set to 4000. The resulting patterns were similar to those shown in Table 4 and Table 5, with an optimum DEM focal distance of around 5 km and a slightly larger elevation scaling of around 5000. There was little difference between the performance with these two elevation scales. All the remaining analyses were completed on the data with false zeroes removed, using the initially determined 5 km DEM focal distance and elevation scaling of 4000.

The impact of including the DEM as an independent variable was further quantified in Table 6. It shows that, compared to the bivariate analysis, the optimal trivariate analysis reduced the MAPE by about 4% for light rainfall days and by about 2% for medium to heavy rainfall days. The trivariate analysis reduced the MAR by about 16% across all days.

**Table 4.** Performance of the interpolation model with different elevation transformation parameters and elevation smoothing scales for light rain days (0-1 mmh$^{-1}$). The minimum values of the RTGCV are shown in bold.

| $a$ | 1 km | 2 km | 3 km | 4 km | 5 km | 6 km | 7 km | 8 km | 9 km | 10 km |
|---|---|---|---|---|---|---|---|---|---|---|
| 1000 | 0.2003 | 0.2005 | 0.1993 | 0.1984 | 0.1981 | 0.1980 | 0.1978 | 0.1978 | 0.1976 | 0.1978 |
| 2000 | 0.1983 | 0.1978 | 0.1981 | 0.1976 | 0.1976 | 0.1973 | 0.1970 | 0.1969 | 0.1969 | **0.1968** |
| 3000 | 0.1975 | 0.1978 | 0.1976 | 0.1974 | 0.1973 | 0.1973 | 0.1973 | 0.1972 | 0.1971 | **0.1967** |
| 4000 | 0.1975 | 0.1976 | 0.1975 | 0.1973 | 0.1972 | 0.1974 | 0.1973 | 0.1970 | 0.1971 | 0.1971 |
| 5000 | 0.1976 | 0.1974 | 0.1973 | 0.1972 | 0.1972 | 0.1972 | 0.1971 | 0.1970 | 0.1971 | **0.1969** |
| 6000 | 0.1975 | 0.1973 | 0.1973 | 0.1972 | 0.1972 | 0.1972 | 0.1970 | 0.1970 | **0.1969** | **0.1969** |
| 7000 | 0.1975 | 0.1973 | 0.1973 | 0.1972 | 0.1972 | 0.1971 | 0.1970 | 0.1970 | **0.1969** | **0.1969** |
| 8000 | 0.1974 | 0.1973 | 0.1972 | 0.1973 | 0.1972 | 0.1971 | 0.1970 | 0.1970 | **0.1969** | 0.1970 |
| 9000 | 0.1975 | 0.1972 | 0.1974 | 0.1972 | 0.1972 | 0.1971 | 0.1970 | 0.1970 | **0.1969** | 0.1970 |
| 10,000 | 0.1974 | 0.1972 | 0.1973 | 0.1972 | 0.1972 | 0.1971 | 0.1970 | 0.1970 | **0.1969** | 0.1970 |

**Table 5.** Performance of the interpolation model with different elevation transformation parameters and elevation smoothing scales for medium to high rain days (> 1 mmh$^{-1}$). The minimum value of the RTGCV is shown in bold.

| a | 1 km | 2 km | 3 km | 4 km | 5 km | 6 km | 7 km | 8 km | 9 km | 10 km |
|---|---|---|---|---|---|---|---|---|---|---|
| 1000 | 0.5536 | 0.5518 | 0.5485 | 0.5449 | 0.5427 | 0.5438 | 0.5431 | 0.5436 | 0.5423 | 0.5442 |
| 2000 | 0.5429 | 0.5408 | 0.5411 | 0.5385 | 0.5372 | 0.5362 | 0.5374 | 0.5374 | 0.5366 | 0.5403 |
| 3000 | 0.5387 | 0.5393 | 0.5377 | 0.5364 | 0.5359 | 0.5352 | 0.5370 | 0.5370 | 0.5376 | 0.5366 |
| 4000 | 0.5387 | 0.5372 | 0.5366 | 0.5357 | **0.5348** | 0.5359 | 0.5363 | 0.5361 | 0.5369 | 0.5362 |
| 5000 | 0.5369 | 0.5366 | 0.5362 | 0.5351 | 0.5356 | 0.5357 | 0.5362 | 0.5464 | 0.5367 | 0.5359 |
| 6000 | 0.5368 | 0.5356 | 0.5351 | 0.5349 | 0.5359 | 0.5364 | 0.5363 | 0.5465 | 0.5363 | 0.5361 |
| 7000 | 0.5358 | 0.5355 | 0.5351 | 0.5350 | 0.5359 | 0.5364 | 0.5363 | 0.5366 | 0.5362 | 0.5360 |
| 8000 | 0.5356 | 0.5354 | 0.5352 | 0.5354 | 0.5362 | 0.5363 | 0.5363 | 0.5366 | 0.5363 | 0.5360 |
| 9000 | 0.5354 | 0.5354 | 0.5356 | 0.5364 | 0.5367 | 0.5464 | 0.5363 | 0.5365 | 0.5358 | 0.5359 |
| 10,000 | 0.5354 | 0.5353 | 0.5361 | 0.5364 | 0.5365 | 0.5461 | 0.5366 | 0.5366 | 0.5359 | 0.5359 |

**Table 6.** Comparison between bivariate and optimal trivariate analyses on light (0-1 mmh$^{-1}$) and medium to high rainfalls (>1 mmh$^{-1}$).

| | Bivariate | | Trivariate | |
|---|---|---|---|---|
| Average rainfall | MAPE | MAR | MAPE | MAR |
| 0-1 mmh$^{-1}$ | 0.0884 | 0.0505 | 0.0851 | 0.0420 |
| >1 mmh$^{-1}$ | 0.9007 | 0.4378 | 0.8816 | 0.3681 |

## 3.3 Temporal evaluation

The hourly time series at 8 hourly stations were extracted from the gridded datasets and compared with the point measurements for the 2017 (Table 7) and 2022 flood events (Appendix D). The CHRain dataset outperforms the hourly BARRA-SY and radar datasets in representing the measured rainfall data, as indicated by the high correlation coefficient of 0.949, compared to 0.234 and 0.154 for BARRA-SY and radar datasets, respectively (Table 7). Note that as the hourly data from the 8 stations were used to generate the CHRain dataset, it is expected that the CHRain can adequately match the hourly rainfall patterns from the measurements. However, it is not necessary for the thin-plate spline interpolation model to generate exact values of rainfall at the gauges. The rainfall value of a grid cell is calculated and smoothed in relation to the rainfall values measured at surrounding gauges.

All the gridded datasets underestimate the hourly measurements, shown by the negative Bias values. The hourly rainfall patterns of the BARRA-SY did not closely reproduce the point data, as suggested by a low correlation coefficient of 0.234 and a negative NSE of -0.493 (Table 7). The discrepancies between the peaks of BARRA-SY and the measured rainfall are also observed in Fig. 2. In all 8 hourly stations, the peaks of the BARRA-SY data are earlier than the peaks in the point measurements. However, the differences in the peak arrival time between the two datasets are not consistent across the 8 hourly gauges, varying from 5 hours at station H058180 to 9 hours at station H058162. The BARRA-SY data also shows an unreasonably high value of rainfall at station H058162 shown in (Fig. 2), compared to other gridded datasets. The performance of the BARRA-SY dataset is even poorer than the hourly disaggregated ANUClimate and AGCD data. Although Acharya et al. (2019) indicated that the average annual rainfall from the BARRA dataset agreed well with the AGCD dataset, our results demonstrate that at the hourly scale the reanalysed data do not reproduce well the variation of rainfall patterns in the Richmond catchment, during high flood events like in 2017.

Compared with other gridded datasets, the hourly radar-derived rainfall data are the least adequate in reproducing the point measurements, observed in both the 2017 and 2022 flood events. The mismatches between radar rainfall data and point measurements were mentioned in previous studies (McMillan et al., 2011; Seo and Krajewski, 2011; Mandapaka et al., 2009; Schleiss et al., 2020). From our analysis, the hourly peak rainfall values from the radar data are 3-20 hours earlier than the peaks measured at the hourly gauges, observed in all 8 validated stations (Fig. 2). The radar dataset has the biggest MAE values in both 2017 and 2022 events compared with other gridded datasets. It is noted that the radar rainfall captures the rainfall in the atmosphere instead of the point measurements on the ground. Therefore, the arrival times of the peaks measured by radar are expected to be earlier than at the rainfall stations. Moreover, the rainfall amounts that reach the ground are affected by winds and vertical variability of rainfall (Schleiss et al., 2020). More analyses need to be done on the pre-processing of the radar dataset before using it for detailed hydrological applications.

A similar analysis on the 24-hour total CHRain data was undertaken. The daily data at 8 different daily gauges, which were not used to generate the CHRain dataset, were extracted for all the gridded datasets. Since the data at the 8 daily gauges were included in constructing the ANUClimate and AGCD datasets, these datasets show better matches to the measurements than the CHRain dataset (Table 8). The 24-hour total rainfall from the CHRain is strongly associated with the daily measurements,

**Table 7.** Evaluation metrics for hourly rainfall extracted from the gridded datasets during the flood event in 2017 at 8 hourly gauges.

|  | Bias | MAE | $r$ | NSE |
|---|---|---|---|---|
| CHRain | -0.600 | 0.861 | 0.949 | 0.866 |
| BARRA-SY | -2.224 | 3.171 | 0.234 | -0.493 |
| Radar | -2.155 | 3.186 | 0.154 | -0.268 |
| ANUClimate | -1.519 | 2.396 | 0.503 | 0.186 |
| AGCD | -1.486 | 2.412 | 0.500 | 0.181 |

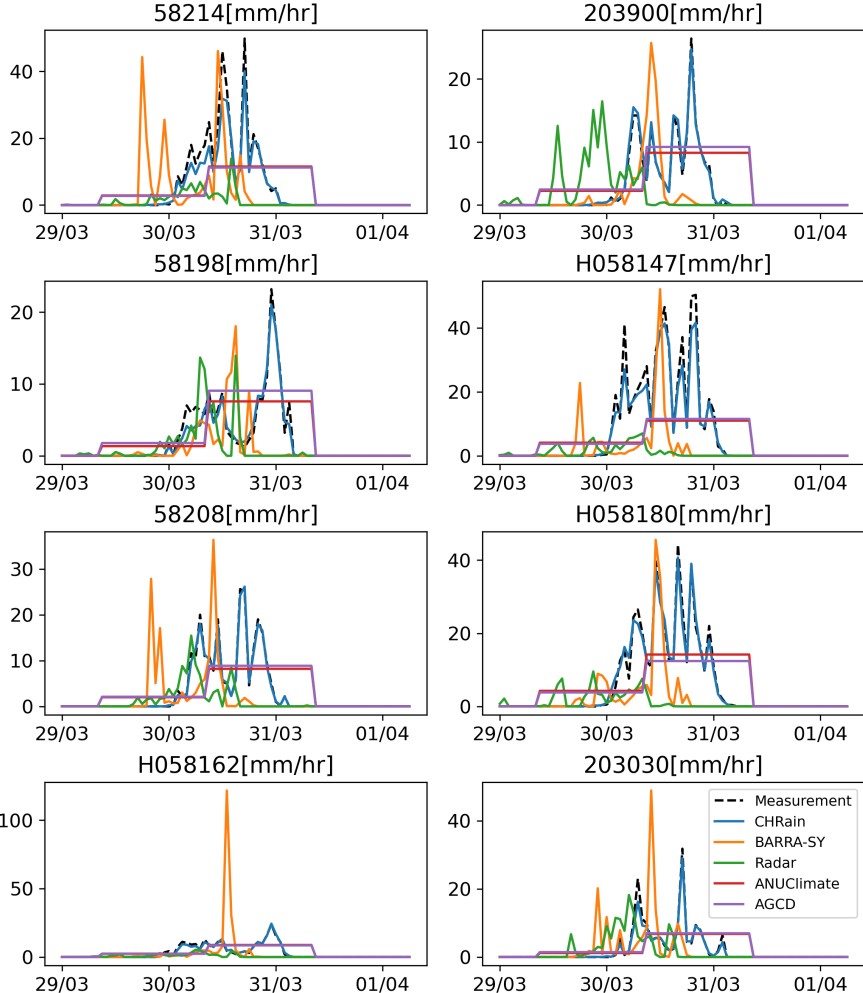

**Figure 2.** Comparison of hourly rainfall data extracted from the gridded datasets at 8 hourly stations during the flood event in 2017.

as indicated by the correlation coefficients of 0.935 in the 2017 flood event and 0.938 in the 2022 flood event. Fig. 3 also demonstrates a good agreement in the peak times between the CHRain, ANUClimate, and AGCD datasets with the daily measurement. The evaluation for the 2022 flood event also resulted in the same conclusion (Appendix D). These results indicate that the CHRain dataset can reproduce the rainfall patterns reasonably well, both at hourly or daily time scales, even at locations without input hourly measurements.

**Table 8.** Evaluation metrics for daily rainfall during the flood event in 2017 at 8 daily gauges.

|            | Bias   | MAE    | $r$   | NSE    |
|------------|--------|--------|-------|--------|
| CHRain     | -5.769 | 8.09   | 0.935 | 0.747  |
| BARRA      | -5.482 | 17.340 | 0.555 | -0.873 |
| Radar      | -6.323 | 16.743 | 0.297 | -0.234 |
| ANUClimate | -1.360 | 4.426  | 0.975 | 0.927  |
| AGCD       | -0.632 | 5.484  | 0.957 | 0.878  |

We also conducted a comparison of the 24 hour total CHRain performance with the daily measurements for the whole period from 2007 to 2022 at 169 daily gauges, which were not included in the generation of CHRain splines. Overall, the CHRain dataset is highly correlated with the daily measurement, indicated by an averaged correlation coefficient of 0.86. Fig. 4 compares the relationship between the 24-hour total CHRain and the daily measurements at 8 selected daily gauges, during days with light rainfall, and medium to extremely heavy rainfall. The CHRain dataset performs better during periods of medium to very heavy rain compared to days with light rain, except at station 58015. For the Richmond River catchment, the light rain events usually occur at a small scale. A slight difference in the locations where rainfall values are extracted from the 24-hour total CHRain splines and the exact locations of daily rainfall gauges can lead to significant variations between the two datasets during light rain periods.

The performance of the CHRain dataset at 169 evaluated daily gauges depends on the distances to the nearest input hourly stations and the density of input gauges around them. The relationship between the correlation coefficients of the 24-hour CHRain and the distance to the nearest input hourly gauge is weak (Fig. 5A). However, the CHRain dataset's performance decreases as the distance from the nearest input gauge increases. Fig. 5B illustrates that the 24-hour total CHRain has a better agreement with the point measurements where the distribution of the input hourly stations is denser. The performance scores spread in a larger range if the gauge density is less than 5 stations per 25 km radius. This is to be expected as the splines are dependent on the available input gauges to fit the rainfall surfaces and as the distance from a input gauge increases the spline is purely the fitted surface without any actual measurement constraint.

### 3.4 Spatial evaluation

From the temporal analysis in Section 3.3, the ANUClimate dataset gives the best match to the daily measurements. In this spatial analysis, we compared the splines from 1 km CHRain dataset to the 1 km ANUClimate and 5 km AGCD datasets. Table

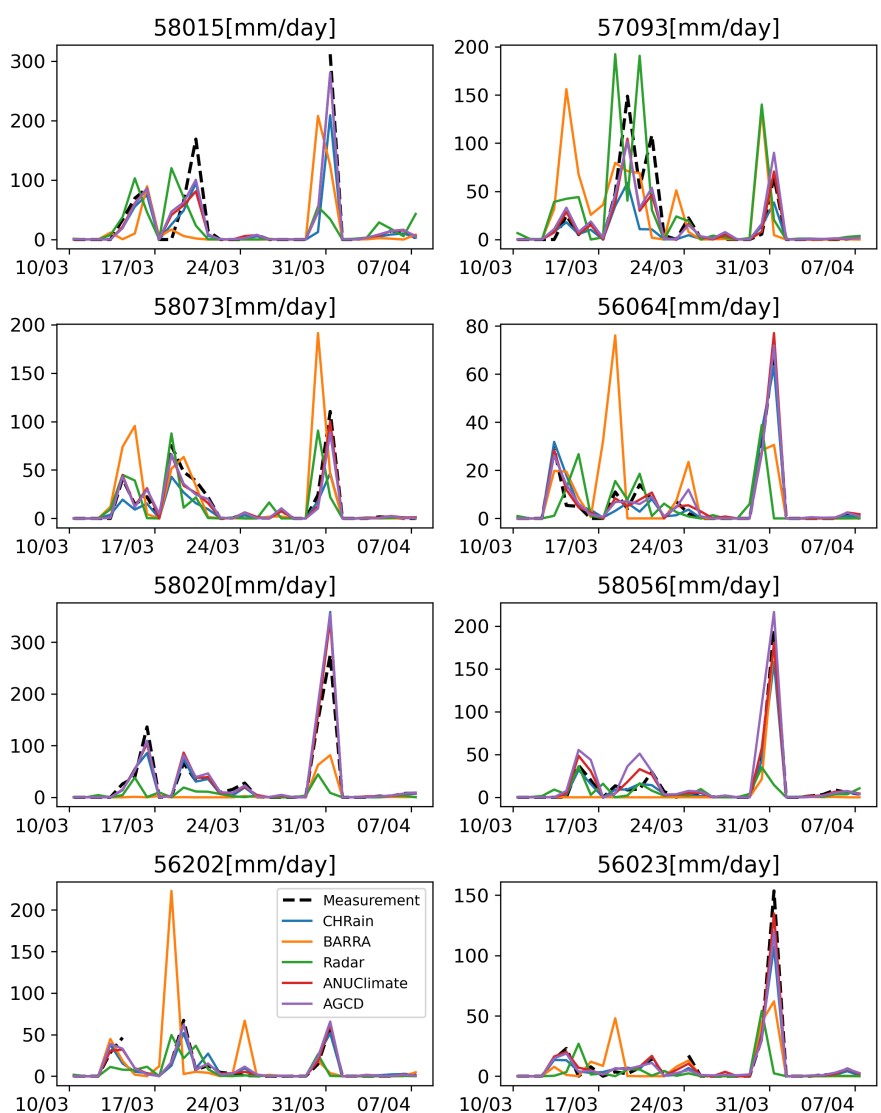

**Figure 3.** Comparison of daily rainfall data extracted from the gridded datasets at 8 daily stations during the flood event in 2017.

9 shows the comparison between the 24-hour total CHRain dataset and the ANUClimate dataset during the 2017 flood event, for the days with heavy rainfall (i.e., the maximum rainfall value in a grid is higher than the 95th percentile).

The averaged Bias score of 0.916 indicates that the 24-hour total CHRain slightly overestimates the wet areas compared with the ANUClimate grids (Table 9). However, the Hit Rate and CSI scores close to 1 demonstrate the high similarity between the two datasets, especially during the extremely high rainfall days on 30-31/03/2017. The evaluation scores increase when the mean rainfall values across the catchment increase. In the days with lighter rain (i.e., lower mean rainfall values), the rains

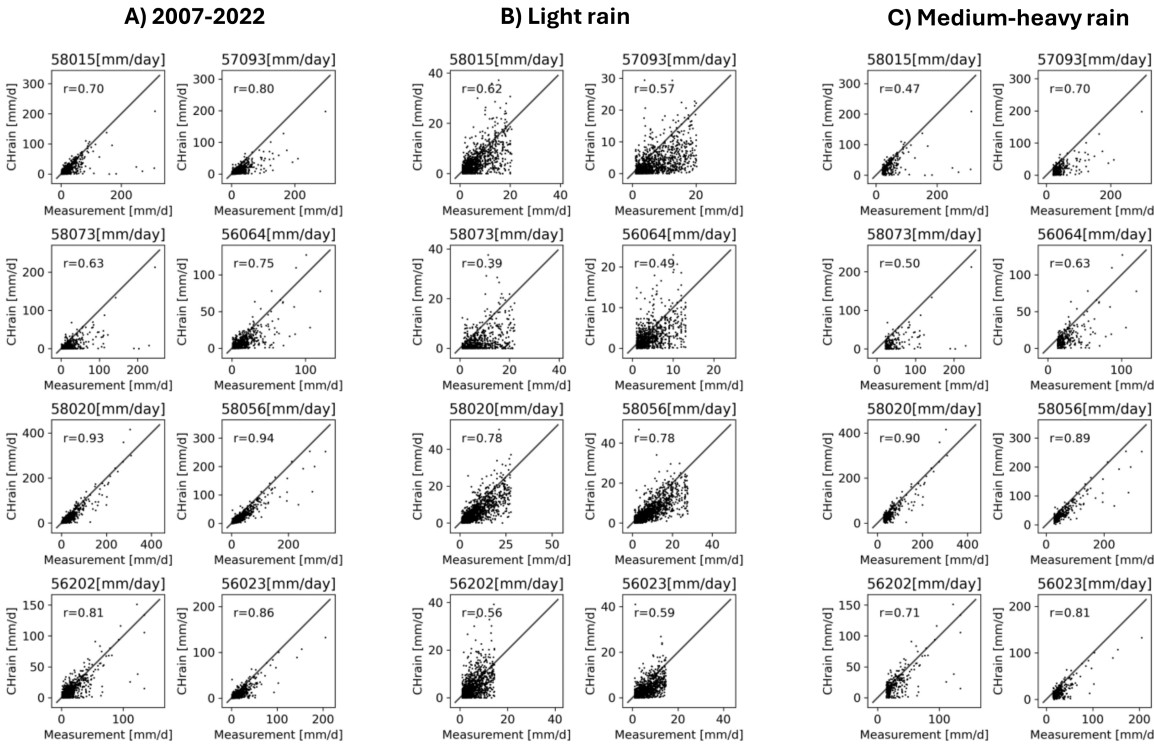

**Figure 4.** Comparison between 24 hour total CHRain and daily point measurements at 8 daily rainfall stations for the whole period from 2007-2022 (Fig. A). Fig. B shows the relationship between the two datasets in light rain days, and Fig. C show the relationship in medium to heavy rain days. $r$ is the correlation coefficient between the two datasets.

usually occur locally and are spread across smaller areas. A small mismatch between the two datasets results in a bigger penalty in the evaluation indices and vice versa.

Even though the spatial resolution of CHRain and ANUClimate datasets is both 1 km (i.e., the 1 km resolution smoothed DEM with focal distance of 5 km), there are bigger variations in the rainfall values in the 24-hour total CHRain splines than in the ANUClimate splines. The difference between the average mean rainfall and the average maximum value of the CHRain

spreads wider from 22.3 $mmd^{-1}$ to 118.8 $mmd^{-1}$, while this range for the ANUClimate is from 26.6 $mmd^{-1}$ to 102.6 $mmd^{-1}$ (Table 9). Fig. 6 compares the rainfall surfaces from the 24-hour total CHRain, the ANUClimate, and the AGCD datasets at the peak of the 2017 flood event on 31/03/2017. There is an agreement in the distribution of the rainfall represented in the three datasets. The variation in the rainfall values within a 5 km window clearly shows that the CHRain can capture the sub-grid variability better than the other 2 datasets with the range of 55 $mmd^{-1}$, 7.4 $mmd^{-1}$, and 0 $mmd^{-1}$ for CHRain, ANUClimate and

AGCD datasets respectively. Interpolating rainfall surfaces using hourly data helps to maintain the details of rainfall distribution in generating the splines compared with using daily data. The local analysis of a specific study area in the CHRain dataset also

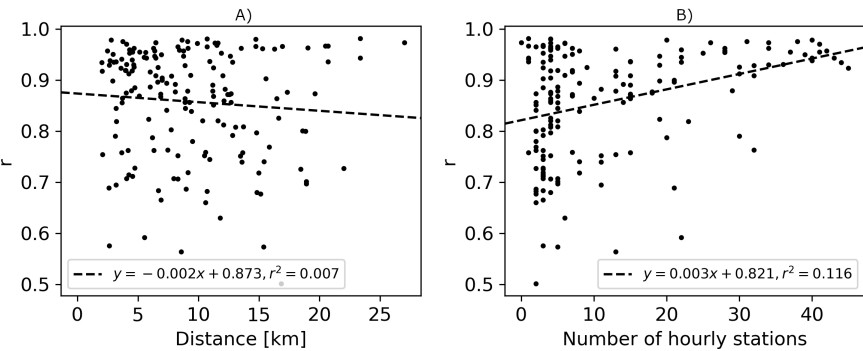

**Figure 5.** A) Relationships between the correlation coefficients (r) of the 24 hour total CHRain and the distance to the nearest input hourly gauge, and B) the correlation coefficients (r) of the 24 hour total CHRain as a function of the hourly gauges density (number of hourly gauges within 25 km radius from a daily station).

increases the influence of topography on the rainfall surface, compared with the analysis for the whole of Australia as the ANUClimate dataset.

In the hourly measurements, the magnitude of the rainfall at each station and the differences between stations are smaller than in the daily data. If the rainfall at one gauge is lower than at other gauges around it, the difference in the magnitude of hourly data is not significant so the spline can "bend" and match the rainfall input at the gauges. In the daily dataset, the differences in rainfall values between stations are bigger since the hourly values are accumulated over 24 hours to daily data. In this case, the smoothing spline interpolation method tries to compensate and balance the rainfall values between stations. Therefore, the smoothing effects are more pronounced in the daily splines compared to the 24-hour total CHRain grid (Fig. 6). This finding also explains the larger variation in the rainfall values in the 24-hour total CHRain dataset compared with the ANUClimate dataset, as shown in Table 9. The rainfall surfaces generated using hourly data can reproduce more details about the rainfall variation and capture the high rainfall values better than the daily splines. On the other hand, it is noted that the hourly splines are more sensitive to the bad zeros in the hourly input dataset. The analysis to generate hourly splines without flagging bad zeros showed some local dipping points in the rainfall surfaces. Including the rainfall occurrence analysis to remove those bad zeros effectively helps to remove those low rainfall areas in the splines.

The rainfall variability at hourly time step during the peak of the 2017 flood event (30-31/03/2017) is presented in Fig. 7. The maximum 24-hour total rainfalls are 225.3 and 487.3 mmd$^{-1}$ on 30 and 31/03/2017, respectively, which were classified as an extremely high rainfall event. The hourly pattern was unevenly distributed, with significant changes occurring both over time and across different locations. The rain started from 1:00 am on 30/03/2017 and reached the peak of 88.5 mmh$^{-1}$ at 11:00 pm on 31/03/2017. The rain stopped 4 hours after reaching the peak. The hourly spatial pattern also shows the movement of the rain front, which moved from the north to the south coast but mostly concentrated towards the northeast boundary of the Richmond River catchment. The spatial distribution and the movement of the rainfall in the CHRain splines contribute to explaining the creation of the high flood event in the Richmond River catchment in 2017. For many hydrological applications such as

**Table 9.** Comparison between 24-hour total CHRain and ANUClimate data during the 2017 flood event.

| Time | Bias | Hit Rate | CSI | MAE [mmd$^{-1}$] | CHRain | | ANUClimate | |
|---|---|---|---|---|---|---|---|---|
| | | | | | Max [mmd$^{-1}$] | Mean [mmd$^{-1}$] | Max [mmd$^{-1}$] | Mean [mmd$^{-1}$] |
| 1/03/2017 | 0.911 | 0.901 | 0.893 | 4.2 | 43.3 | 4.1 | 28.6 | 7.4 |
| 2/03/2017 | 0.728 | 0.721 | 0.715 | 4.5 | 35.9 | 3.7 | 37.0 | 8.0 |
| 3/03/2017 | 0.554 | 0.498 | 0.472 | 2.6 | 82.9 | 3.1 | 35.2 | 2.3 |
| 5/03/2017 | 0.805 | 0.800 | 0.797 | 4.0 | 58.1 | 8.8 | 51.5 | 10.4 |
| 6/03/2017 | 0.667 | 0.657 | 0.650 | 2.9 | 27.7 | 2.1 | 29.3 | 4.7 |
| 13/03/2017 | 1.154 | 0.972 | 0.823 | 2.1 | 44.0 | 9.4 | 45.4 | 9.2 |
| 14/03/2017 | 1.005 | 0.998 | 0.992 | 7.1 | 43.0 | 12.3 | 51.6 | 18.9 |
| 15/03/2017 | 0.998 | 0.992 | 0.986 | 5.4 | 176.5 | 19.4 | 102.0 | 22.0 |
| 16/03/2017 | 0.959 | 0.954 | 0.949 | 9.8 | 131.6 | 27.2 | 146.2 | 36.1 |
| 18/03/2017 | 0.948 | 0.940 | 0.933 | 8.1 | 160.9 | 24.1 | 146.4 | 29.3 |
| 19/03/2017 | 1.050 | 0.993 | 0.939 | 11.6 | 196.3 | 26.0 | 141.7 | 35.0 |
| 20/03/2017 | 1.010 | 0.995 | 0.981 | 8.9 | 131.7 | 16.6 | 81.1 | 23.1 |
| 21/03/2017 | 0.980 | 0.975 | 0.969 | 7.0 | 103.6 | 23.5 | 91.7 | 26.9 |
| 24/03/2017 | 0.923 | 0.909 | 0.896 | 4.4 | 53.8 | 8.6 | 37.6 | 10.2 |
| 30/03/2017 | 1.006 | 1.000 | 0.994 | 10.1 | 225.3 | 50.7 | 266.2 | 47.3 |
| 31/03/2017 | 1.005 | 1.000 | 0.994 | 22.8 | 487.3 | 135.1 | 428.1 | 155.3 |
| 6/04/2017 | 0.864 | 0.855 | 0.847 | 2.0 | 17.6 | 4.1 | 23.7 | 5.9 |
| Average | 0.916 | 0.892 | 0.872 | 6.9 | 118.8 | 22.3 | 102.6 | 26.6 |

simulating the flow in small river channels, the variation of rainfall patterns is essential to correctly estimate the accumulated
volumes and arrival times of floods in rapid responding catchments (Acharya et al., 2022; Lewis et al., 2018; Lerat et al., 2022).

## 4   Discussion

Compared to daily or monthly data, the hourly data contains significantly more zeros, which can increase the instability of the
interpolation model. This paper is the first to test the ability of the ANUSPLIN program to generate hourly rainfall surfaces. It
has also incorporated a robust automated process to remove false zeros from the data. False zeros are a very common problem
with rainfall observations. They are hard to detect by applying simple thresholds. Comparison with hourly radar rainfall data
indicates that the spatial occurrence based corrections are highly reliable with an accuracy of up to around 98%. The method
proposed in this study has been successfully applied to generate a 1 km hourly gridded rainfall dataset for a larger area. Hourly
rainfall data are essential for many hydrological, ecological, and meteorological applications (Lewis et al., 2018; Hatono et al.,
2022).

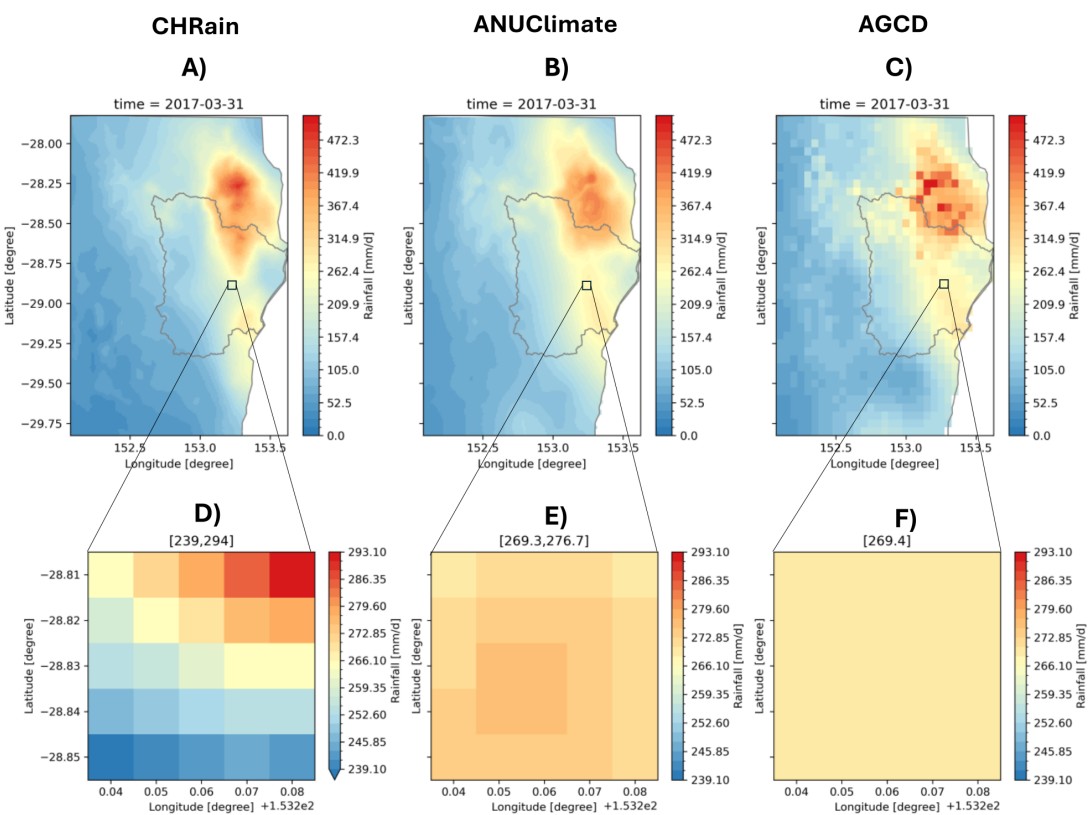

**Figure 6.** Comparison between CHRain, ANUClimate and AGCD datasets on 31/03/2017. A, B, and C show the rainfall surfaces from three datasets for the whole study area. D, E, and F show the 5-km areas at the hourly gauge 58214.

Including elevation data enhances the performance of the thin-spline interpolation model in generating hourly rainfall surfaces, more significantly during larger rainfalls. While the response of the splines to the topography during light rain days is quite broad, the elevation data has greater impacts during larger rain days and results in the clear optimal values for the DEM transformation parameter and the smoothing distance. There are higher resolution DEMs than the 1 km used in the analysis in this paper. However, the result suggests including finer topographic data does not result in better rainfall surfaces at higher

spatial resolution. For our study area, the optimal values for the transformation parameter $a$ and the DEM focal distance are around 4000 to 5000 and 5 km, respectively. The optimal DEM focal distance of 5 km is in agreement with the analysis of Sharples et al. (2005), who showed that similarly averaged DEMs with focal distances from 5 to 10 km performed best in interpolating monthly rainfall across Australia. On the other hand, the optimal elevation scaling of around 4000 to 5000 corresponds to a vertical exaggeration of around 20. This is somewhat less than the vertical exaggeration of around 100 found

with spatial analyses of rainfall at broader time scales by Hutchinson (1995) and Johnson et al. (2016). This suggests that

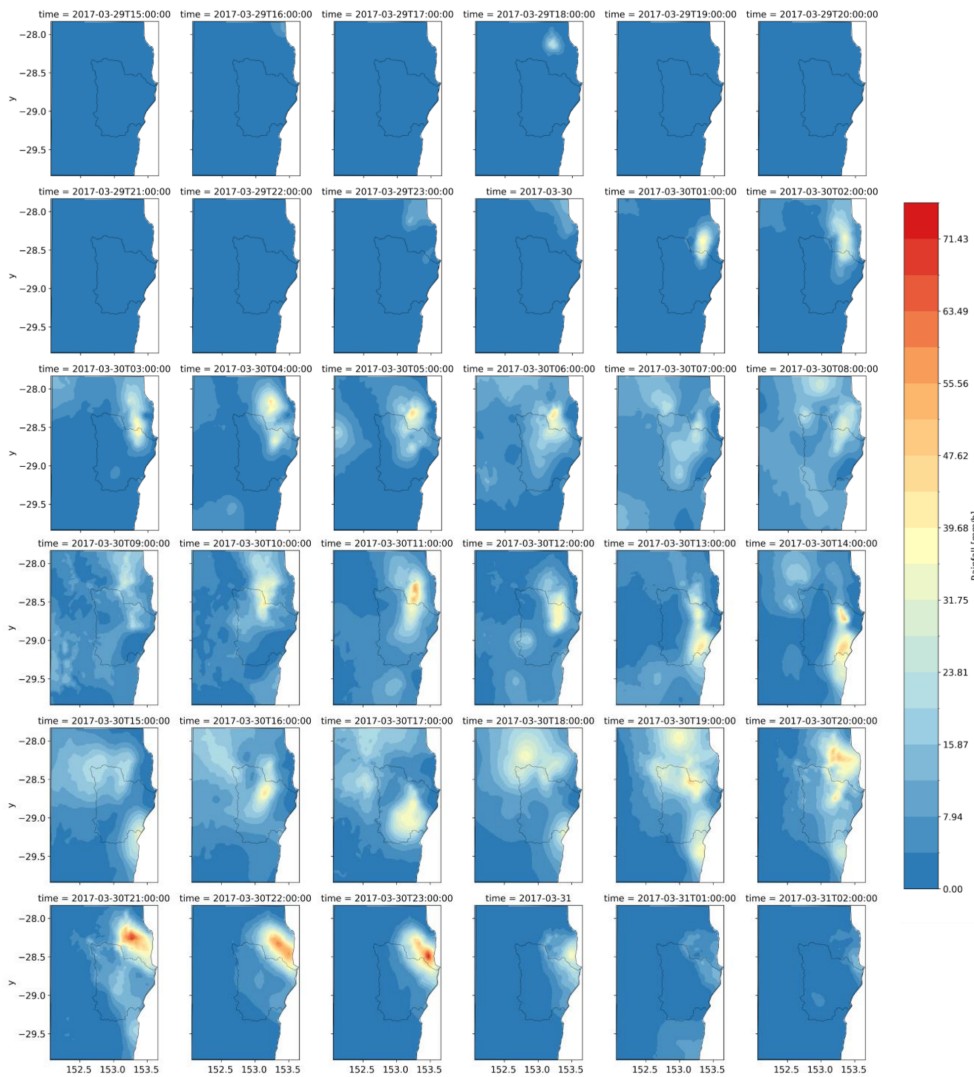

**Figure 7.** Hourly rainfall splines from the CHRain dataset during the peak of the 2017 flood event on 30-31/03/2017.

hourly rainfall, though significantly influenced by elevation, has a less consistent dependence on elevation than rainfall values recorded at broader time scales.

The initial hourly rainfall occurrence analysis appears to have been effective in detecting and removing the many false zeroes that can arise with automatically recorded hourly rainfall data. This was aided by the limited spatial extent of this rainfall analysis. The detections would likely to be less reliable when applied to sites with no relatively near neighbours.

The CHRain dataset most closely aligns with the hourly measurements compared to other datasets, including BARRA-SY and radar data. From our analysis, the reanalysed BARRA-SY data does not reproduce the hourly patterns of the recorded rainfall in the Richmond River catchment, and it performs worse than the daily averaged to hourly datasets (e.g., from ANU-

Climate and AGCD data). The results from our analysis disagrees with the conclusion by Acharya et al. (2022), showing that using the hourly patterns from the BARRA dataset is useful to disaggregate the daily AGCD data to hourly for rainfall-runoff modelling. Rhodes et al. (2015) also concluded that the reanalysed products can only capture 40-65% wet areas during extreme rainfall events in the UK and Wales. The objective of generating the reanalysed datasets (e.g., BARRA) is to provide consistent information of historical climate variations including precipitation at a higher temporal scale (hourly), especially when and where the measurement data are not available. The datasets are valuable for climatological studies across a much larger area and longer periods. Currently, the reanalysed data did not consider the point measurements in the generation process (Su et al., 2019). Therefore, the reanalysed rainfall data are not yet suitable for using in detailed hydrological/hydrodynamic modelling. Further research need to be conducted to address the uncertainties in reanalysis data and enhance its precision for using in modelling applications.

The method to generate 1-km resolution hourly rainfall data presented in this study opens an opportunity to produce high spatiotemporal accurate rainfall datasets for areas where detailed modelling is required in Australia, and where hourly measurements are available. The ANUSPLIN program has options to incorporate spatially dependent variables, such as rainfall observations from satellites or radars. However, because of the artifacts, there are limitations in using hourly rainfall extracted from radar datasets (McMillan et al., 2011; Schleiss et al., 2020). We also expect the radar estimates of rainfall to improve over time as it is still a developing technology and there will be major advances in this field with time. As for now, for future studies, we suggest investigating the relationships between the radar observations and the ground measurements. Then, we can utilize the distribution of rainfall intensity in radar datasets for interpolating rainfall splines.

The reliability of the CHRain dataset depends on the intensity of an event, the quality of the input hourly data at rainfall stations, the distribution of the hourly gauges in the area of interest, and the distances of the point/area of interest to the nearest input gauge. Despite the removal of suspicious point measurements through automated quality control and manual checks, errors that fall outside the checking criteria may still exist. Disaggregating daily data into hourly intervals helps to represent hourly rainfall patterns in areas where hourly gauges are scarce. However, this method cannot accurately capture changes in the pattern caused by the movement of the rain front (unless short interval radar images are used to provide this information). The performance of the CHRain is better during the medium to heavy events, and when the rain is spread over a larger area. In general, Ebert (2008) stated that it is more challenging to simulate the light intensity rainfall over a small area. During these events, the model is highly sensitive to the input from rainfall gauges. Small errors in the rainfall record or slight variations in the location of the gauges can lead to significant differences between the generated data and the actual measurements. Considering the computational efficiency of the ANUSPLIN program and the distribution of the hourly rainfall stations, with applications that do not require the observation of rainfall across an extensive area (i.e., for the whole of Australia), we suggest generating splines locally to increase the accuracy and reliability of the rainfall surfaces.

The spatial analysis proves that the 1 km 24-hour total CHRain dataset can show more detail in the rainfall variation than in the 1 km daily ANUClimate dataset. The hourly CHRain splines also demonstrate the movement and distribution of the rainfall across the Richmond River catchment. This information is essential for understanding and accurately modelling large flood events (Davis, 2001; Westra et al., 2014). As always with coastal storm fronts, these are fast moving storm fronts and the

total daily rainfall may only fall within a couple of hours of the day with hardly any or no rainfall after the front has passed over the area of interest. This creates a major limitation in floodplain inundation modelling as this lumped daily representation of rainfall does not provide the model with the necessary inputs and this could lead to major differences in peak heights and timing. However, the hourly splines are more sensitive to the accuracy of input data, including the DEM and the measured rainfall inputs. To apply the thin-plate spline interpolation method on larger areas (e.g., for the whole of Australia), thorough investigations need to be undertaken on the quality control of the hourly measurements to minimise spatial-temporal errors of gauged data (Lewis et al., 2018; Tang et al., 2018).

## 5 Conclusions

This paper has examined the topographic dependence of hourly rainfall patterns. It has found that higher rainfalls have a consistent dependence on DEM parameters, with an optimal spatial resolution of around 5 km, consistent with previous studies, and a reduced exaggeration of elevation dependence compared to previous studies of daily and monthly rainfall.

This paper introduced a method to generate hourly 1 km resolution gridded rainfall data, that are suitable for hydrological/hydrodynamic modelling applications. The temporal analysis demonstrated that the CHRain dataset is highly correlated with the rainfall measurements at both hourly and daily time steps (with correlation coefficients of 0.949 and 0.935, relatively). The spatial evaluation indicated that the CHRain outperforms the ANUClimate and AGCD datasets, which are the most commonly used reliable rainfall datasets in Australia, in representing the 5 km sub-grid rainfall distribution at the Richmond River catchment. The 24-hour total CHRain dataset can also capture high rainfall values better than the ANUClimate dataset (e.g., Bias = 0.916). The hourly CHRain surfaces can capture the movement of rain fronts and the dynamic temporal variations of the rainfall during heavy rainfall events. Those rainfall characteristics are required to achieve more accurate flood simulation/modelling.

The reliability of the proposed method depends on various factors, such as the event rainfall intensity, quality of input hourly data, distribution and proximity of rainfall stations, and the process of disaggregating daily data into hourly intervals. For future studies, we suggest investigating the inclusion of rainfall intensity from radar patterns into the thin-spline interpolation, applying a thorough quality control, and utilising a more advanced disaggregation method to increase the reliability of the CHRain dataset.

## Appendix A: Evaluation metrics and indices

The bias, Mean Absolute Error (MAE), correlation coefficient ($r$), and Nash–Sutcliffe Efficiency (NSE) metrics are calculated in the temporal evaluation.

$$\text{Bias} = \frac{\sum_{i=1}^{N}(\hat{Y}_i - Y_i)}{N}, \tag{A1}$$

$$\text{MAE} = \frac{\sum_{i=1}^{N}(|\hat{Y}_i - Y_i|)}{N}, \tag{A2}$$

$$r = \frac{\sum_{i=1}^{N}(Y_i - \mu_{Y_i})(\hat{Y}_i - \mu_{\hat{Y}_i})}{\sqrt{\sum_{i=1}^{N}(Y_i - \mu_{Y_i})^2 \sum_{i=1}^{N}(\hat{Y}_i - \mu_{\hat{Y}_i})^2}}, \tag{A3}$$

$$\text{NSE} = 1 - \frac{\sum_{i=1}^{N}(\hat{Y}_i - Y_i)^2}{\sum_{i=1}^{N}(Y_i - \mu_{Y_i})^2}, \tag{A4}$$

where $\hat{Y}_i$ is the predicted rainfall, $Y_i$ is the measured rainfall, $\mu_{\hat{Y}_i}$ is the mean of predicted rainfall, $\mu_{Y_i}$ is the mean of measured rainfall, $r$ is the correlation coefficient between modeled and predicted rainfall, $\sigma_{\hat{Y}_i}$ is the standard deviation of predicted rainfall, $\sigma_{Y_i}$ is the standard deviation of measured rainfall and $N$ is the total number of observations.

For the spatial analysis, we used Bias, Hit Rate, and CSI scores to compares between gridded datasets (Ebert, 2008).

$$\text{Bias} = \frac{\text{hits} + \text{false alarms}}{\text{hits} + \text{misses}}, \tag{A5}$$

$$\text{Hit Rate} = \frac{\text{hits}}{\text{hits} + \text{misses}}, \tag{A6}$$

$$\text{CSI} = \frac{\text{hits}}{\text{hits} + \text{misses} + \text{false alarms}}. \tag{A7}$$

## Appendix B: The spline occurrence analysis for the high rainfall day on 30/03/2017

The 1's in Table B1 denote the false zeroes, as determined by the spline occurrence analysis, over the 24 hours for the high rainfall day on 30/03/2017. On this day almost all sites recorded all positive rainfall data values after the first three hours. Sites H057005, H057123, H058068, H058231 recorded zero values for all 24 hours. Sites H558071, H558076, H558090, 204900

recorded zero values for the first 8 or 9 hours followed by missing data. Site H558082 had 4 zero values over the first 9 hours

followed by missing data. All of these false zero detections appear to be correct. The few remaining isolated detections are at sites with positive rainfall values on preceding or succeeding days.

**Table B1.** The spline occurrence analysis for 30/3/2017

| Site | | | | | | | | | | | | | | | | | | | | | | | | |
|---|---|---|---|---|---|---|---|---|---|---|---|---|---|---|---|---|---|---|---|---|---|---|---|---|
| H056199 | 0 0 0 | 0 0 0 | 0 0 0 | 0 0 0 | 0 0 0 | 0 0 0 | 0 0 0 | 1 0 0 |
| H057005 | 0 0 0 | 1 1 1 | 1 1 1 | 1 1 1 | 1 1 1 | 1 1 1 | 1 1 1 | 1 0 0 |
| H057123 | 0 0 0 | 0 1 1 | 1 1 1 | 0 0 1 | 1 1 1 | 1 1 1 | 1 1 0 | 0 0 0 |
| H058068 | 0 0 0 | 0 1 1 | 1 1 1 | 1 0 1 | 1 1 1 | 1 1 1 | 1 1 1 | 0 0 1 |
| H058231 | 0 0 0 | 0 1 1 | 1 1 1 | 0 0 1 | 1 1 1 | 1 1 1 | 1 1 1 | 1 0 0 |
| H558071 | 1 1 1 | 1 1 1 | 1 1 0 | 0 0 0 | 0 0 0 | 0 0 0 | 0 0 0 | 0 0 0 |
| H558076 | 0 0 1 | 1 1 1 | 1 1 1 | 0 0 1 | 0 0 0 | 0 0 0 | 0 0 0 | 0 0 0 |
| H558082 | 0 1 1 | 0 0 0 | 0 1 1 | 0 0 0 | 0 0 0 | 0 0 0 | 0 0 0 | 0 0 0 |
| H558090 | 1 1 1 | 1 1 1 | 1 1 1 | 0 0 0 | 0 0 0 | 0 0 0 | 0 0 0 | 0 0 0 |
| 041525 | 0 0 0 | 0 0 0 | 0 0 0 | 0 0 0 | 0 0 0 | 0 0 0 | 0 0 1 | 1 0 0 |
| 204403 | 0 0 0 | 0 0 0 | 0 0 0 | 0 0 0 | 0 0 0 | 0 0 0 | 0 0 0 | 0 0 1 |
| 145020A | 0 0 0 | 0 0 0 | 0 0 0 | 0 0 0 | 0 0 0 | 0 0 0 | 0 0 0 | 0 0 1 |
| 145027A | 1 0 0 | 0 0 0 | 0 0 0 | 0 0 0 | 0 0 0 | 0 0 0 | 0 0 0 | 0 0 0 |
| 145003B | 0 0 0 | 0 0 0 | 0 0 0 | 1 0 0 | 0 0 0 | 0 0 0 | 0 0 0 | 0 0 0 |
| 204007 | 0 0 0 | 0 0 0 | 0 0 0 | 0 0 0 | 0 0 0 | 0 0 0 | 0 0 0 | 1 0 0 |
| 204900 | 0 0 0 | 1 1 1 | 1 1 1 | 0 0 0 | 0 0 0 | 0 0 0 | 0 0 0 | 0 0 0 |
| 204033 | 0 0 0 | 0 0 0 | 0 0 0 | 0 0 0 | 0 0 0 | 0 0 0 | 0 0 1 | 1 0 0 |
| 058097 | 0 0 0 | 0 0 0 | 0 0 0 | 0 0 1 | 0 0 0 | 0 0 0 | 0 0 0 | 0 0 0 |
| 058061 | 0 0 0 | 0 0 0 | 0 0 0 | 0 0 1 | 0 0 0 | 0 0 0 | 0 0 0 | 0 0 0 |
| 057003 | 0 0 1 | 0 0 0 | 0 0 0 | 0 0 0 | 0 0 0 | 0 0 0 | 0 0 0 | 0 0 0 |

## Appendix C: Statistics of the hourly measurement data for the flood events in 2017 and 2022

**Table C1.** Statistics for the observed hourly rainfall during the flood event in 2017.

| Station ID | Mean [mmh$^{-1}$] | Max [mmh$^{-1}$] | Std [mmh$^{-1}$] | P$_3$ [%] | P$_6$ [%] | P$_{12}$ [%] | P$_{24}$ [%] |
|---|---|---|---|---|---|---|---|
| 58214 | 9.3 | 41.0 | 11.9 | 92.7 | 82.6 | 67.5 | 50.1 |
| 203900 | 5.7 | 30.2 | 6.8 | 71.3 | 54.6 | 49.8 | 31.0 |
| 58198 | 4.5 | 32.2 | 6.8 | 71.6 | 37.3 | 28.8 | 25.5 |
| H058147 | 13.1 | 83.6 | 17.5 | 67.4 | 44.5 | 39.6 | 35.0 |
| 58208 | 5.6 | 26.0 | 6.3 | 86.9 | 69.5 | 50.4 | 39.7 |
| H058180 | 12.4 | 50.7 | 13.2 | 62.9 | 56.9 | 55.6 | 54.1 |
| H058162 | 7.8 | 33.4 | 8.9 | 81.3 | 47.3 | 38.2 | 37.2 |
| 203030 | 7.5 | 35.8 | 8.2 | 83.6 | 72.6 | 47.9 | 39.7 |
| **Average** | **8.2** | **41.6** | **9.95** | **77.2** | **58.2** | **41.0** | **39.0** |

**Table C2.** Statistics for the observed hourly rainfall during the flood event in 2022.

| Station ID | Mean [mmh$^{-1}$] | Max [mmh$^{-1}$] | Std [mmh$^{-1}$] | P$_3$ [%] | P$_6$ [%] | P$_{12}$ [%] | P$_{24}$ [%] |
|---|---|---|---|---|---|---|---|
| 58214 | 5.3 | 41.0 | 8.7 | 92.7 | 82.6 | 67.5 | 50.1 |
| 203900 | 2.3 | 30.2 | 4.2 | 71.3 | 54.6 | 49.8 | 31.0 |
| 58198 | 3.5 | 93.4 | 7.4 | 46.4 | 28.2 | 21.5 | 14.7 |
| H058147 | 3.7 | 83.6 | 8.3 | 67.4 | 44.5 | 39.6 | 35.0 |
| 58208 | 2.4 | 26.0 | 4.1 | 86.9 | 69.5 | 50.4 | 39.7 |
| H058180 | 3.0 | 50.7 | 6.4 | 62.9 | 56.9 | 55.6 | 54.1 |
| H058162 | 3.2 | 33.4 | 5.5 | 81.3 | 47.3 | 38.2 | 37.2 |
| 203030 | 2.8 | 40.8 | 5.2 | 50.5 | 35.0 | 27.7 | 15.3 |
| **Average** | **3.3** | **49.9** | **6.2** | **69.9** | **52.3** | **43.8** | **34.6** |

# Appendix D: Temporal analysis of the flood event in 2022

**Table D1.** Evaluation metrics for the flood event in 2022, observed at 8 validated hourly gauges.

|  | Bias | MAE | $r$ | NSE |
|---|---|---|---|---|
| CSIROGrid | -0.608 | 1.681 | 0.928 | 0.839 |
| Radar | -4.000 | 6.051 | 0.223 | -0.352 |
| ANUClimate | -2.575 | 4.555 | 0.502 | 0.129 |
| AGCD | -2.623 | 4.576 | 0.496 | 0.113 |

**Table D2.** Evaluation metrics for the daily rainfall during the flood event in 2022 at 8 daily gauges.

|  | Bias | MAE | $r$ | NSE |
|---|---|---|---|---|
| CSIROGrid | -4.713 | 6.908 | 0.938 | 0.800 |
| Radar | -3.790 | 14.950 | 0.690 | 0.134 |
| ANUClimate | -1.825 | 3.724 | 0.988 | 0.964 |
| AGCD | -1.330 | 5.295 | 0.966 | 0.911 |

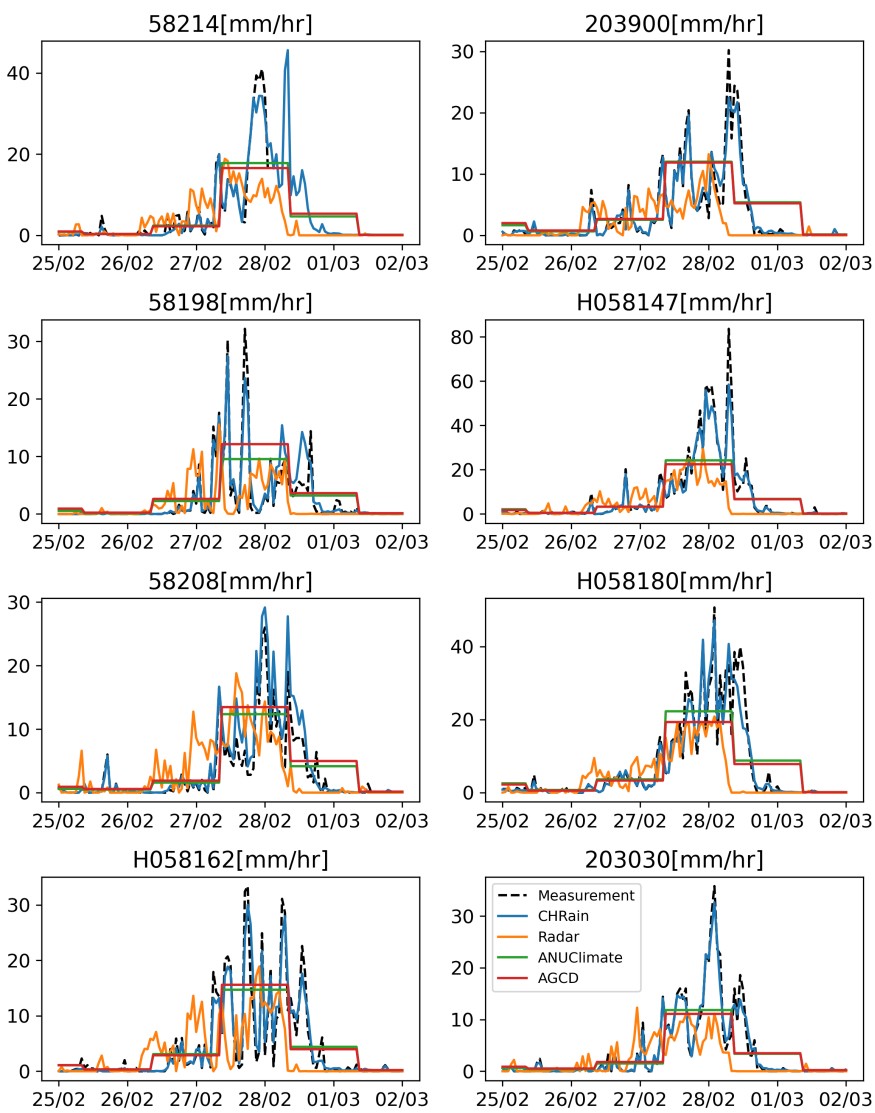

**Figure D1.** Comparison of hourly rainfall data at 8 hourly stations during the flood event in 2022.

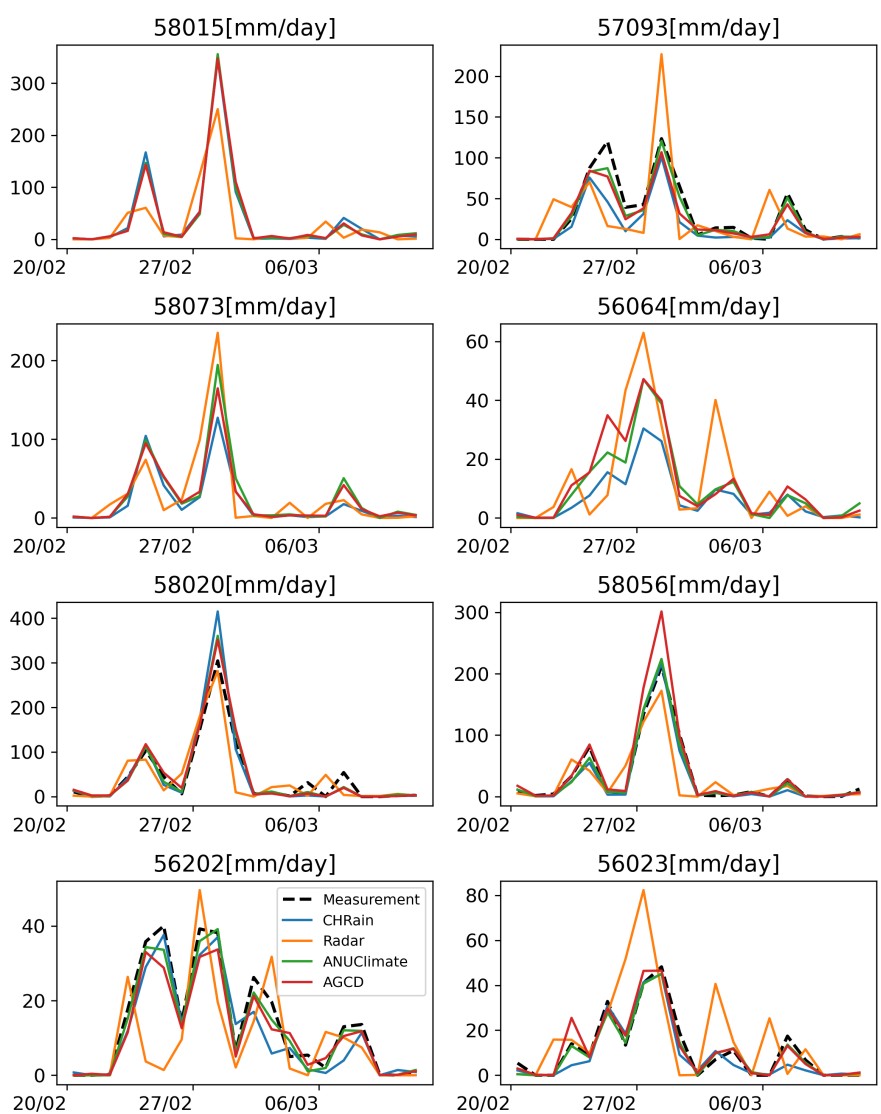

**Figure D2.** Comparison of hourly rainfall data at 8 daily stations during the flood event in 2022.

## Code and data availability

We provided Python scripts and a sample to prepare inputs and generate hourly rainfall splines at 1 km resolution using the ANUSPLIN program at https://doi.org/10.5281/zenodo.17686121. Users will need to contact the Fenner School of Environment & Society (https://fennerschool.anu.edu.au/research/products/anusplin-version-4-4) to get the ANUSPLIN program license.

*Author contributions.* Chi Nguyen developed the CHRain dataset, did the analysis, and prepared the manuscript. Jai Vaze designed the experiments, developed the CHRain dataset, and edited the manuscript. Cherry Mateo developed the CHRain dataset and edited the manuscript. Michael Hutchinson developed the false zeros detection and removal process, including the occurrence analysis with radar rainfall data, and provided guidance through out the study. Jin Teng provided guidance and edited the manuscript.

*Competing interests.* We declare there is no competing interests.

*Acknowledgements.* This work is undertaken as part of the Northern Rivers Resilience Initiative project led by CSIRO and funded by the National Emergency Management Agency (NEMA). We would like to thank the Bureau of Meteorology for providing the measurements at rainfall stations and the radar data. We thank the Rous County Council for providing the hourly rainfall measurements at Rocky Creek Dam and Emigrant Creek Dam during the 2022 flood event, which are critical inputs for this analysis. We would like to thank other team members at CSIRO including Julien Lerat, Bill Wang, Steve Marvanek and Catherine Ticehurst for downloading and preparing the data used in the paper. We appreciate the comments and suggested ideas from all three reviewers. The contributions significantly strengthened the analyses in the final version of the paper.

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
