# Peer review of "Elevation dependent spatial interpolation of hourly rainfall for accurate flood inundation modelling"

_Hydrology and Earth System Sciences, 2024_

## Community Comment (CC1)

[Figure]

[Figure]

**Figure 2.** Comparison of hourly rainfall data extracted from the gridded datasets at 8 hourly stations during the flood event in 2017.

days with light rainfall, and medium to extremely heavy rainfall. The CHRain dataset performs better during periods of medium
305   to very heavy rain compared to days with light rain, except at station 58015. For the Richmond River catchment, the light rain
events usually occur at a small scale. A slight difference in the locations where rainfall values are extracted from the 24-hour
total CHRain splines and the exact locations of daily rainfall gauges can lead to significant variations between the two datasets
during light rain periods.

[Figure]

[Figure]

**Figure 3.** Comparison of daily rainfall data extracted from the gridded datasets at 8 daily stations during the flood event in 2017.

affecting storm depth, intensity, and local climate is topography (orographic effects). Interpolating rainfall surfaces using daily point data also increases the smoothing effect in generating the splines than using the hourly measurements.

335    Fig. 6 compares the rainfall surfaces from the 24-hour total CHRain, the ANUClimate, and the AGCD datasets at the peak of the 2017 flood event on 31/03/2017. It can be seen that there is an agreement in the distribution of the rainfall represented in the three datasets. The variation in the rainfall values within a 5 km window clearly shows that the CHRain can capture the sub-grid variability far better than the other 2 datasets with the range of 57.9 mmd$^{-1}$, 7.4 mmd$^{-1}$, and 0 mmd$^{-1}$ for CHRain,

---

## Community Comment (CC2)

The need for better spatial and temporal resolutions of hydrological data has been an ongoing demand by researchers. To meet the demand, for example, with the advancement of technology and resources, the joint collaboration of NASA and USGS has decided to improve the spatial and temporal resolutions of its Landsat products (e.g., Landsat 8, 9) by 2030 with the impending introduction of Landsat NEXT, which is expected to have a spatial resolution of 10 m for some of the bands. In this manuscript, the authors propose a methodology to generate gridded rainfall data for a river basin in Australia. The spatial and temporal resolutions of the generated data are 1 km and 1 h, respectively. The proposed methodology is based on a thin-plate spline interpolation model that was introduced in 1990. The authors claim that the generated gridded rainfall data has outperformed the other sources of rainfall data available in Australia.

The current version of the manuscript has many flaws. I suggest the authors go through the video on how to write a research paper (https://www.egu.eu/webinars/229/how-to-write-a-research-paper/). Although the presentation is not the best on the topic, I believe that it has something to grasp. Thanks to the UK (University of Leeds), even after Brexit, it is playing the role of gladiator to safeguard the education system of the EU.

1) The title of the manuscript needs to be examined by the journal office. I presume that the journal office has appointed a handling editor and referees after going through their academic transcripts and their PhD works.

There is a need to understand the way titles have been coined in manuscripts in this journal office. For example, recently, in this journal office, the handling editor published a manuscript authored by Google Research. The title of the manuscript is Never Train a Long Short-Term Memory (LSTM) Network on a Single Basin. Is there a verb in the title? The title of this manuscript is "VERY HIGH spatial and temporal resolution rainfall data for accurate flood inundation modeling." Is there a verb in the title? To evaluate the quality of the review process of the journal office, it would be more appropriate to conduct research to examine the titles of the manuscripts that have been published by the journal office since the establishment of the journal.

2) See the attached file(Figure 2_3.pdf).  Are the figures correct? Have you pasted incorrectly? I am expecting the investors of the journal office to spend a few euros to develop an AI/ML-generated model to do a preliminary check on the manuscripts that are submitted to this journal office.

3) Refer to PART II

---

## Community Comment (CC3)

1-3) Refer to PART I

4) Australia is a developed country. If I have misunderstood, do correct me. Therefore, I am expecting the authors to produce their own datasets to map Figure 1. There are too many sources/references cited to produce the map shown in Figure 1. This gives a bad impression about the state of water resources management in Australia. Since the spatial locations of raingages are available from the Bureau of Meteorology, it should be a matter of a few minutes to have the map of gages. Moreover, Australia has a fine resolution DEM. Therefore, producing the catchments and stream networks for the nation is not a pain-taking task. Are the authors lacking knowledge and experience?

5) In the current version of the manuscript, the thresholds that are used to remove unrealistically high hourly and daily rainfall data are not justified. What was the reason to set the daily threshold to 1500 mm/d? What was the reason to set the hourly threshold to 300 mm/h? Considering the authors' statements [LN 171-172, LN 106-107, LN 104] in the current version of the manuscript, these threshold values are not justified.

6) LN 170-Some unusually high values of hourly rainfall, mostly occurring at **midnight**, were detected. LN 168- The suspicious data were removed, including negative, ...

The authors have found some interesting points in the Australian datasets. It would be more appropriate to include a few lines to elaborate more on this for the readers to learn more on what has been found by the authors. What was the reason to have negative values in the dataset? What was the reason to have very high values of hourly rainfall, specifically at midnights? Are these observational errors, or are these due to some interesting unexplored researchable areas?

7) Refer to PART III

---

## Community Comment (CC4)

1-6) Refer to PART I & II

7) As per the authors, the spatial extent of the analysis is 30,389 km2 [LN 111]. Moreover, there are 139 hourly stations for generating hourly rainfall surfaces using the proposed method. This includes the disaggregated stations (i.e., daily to hourly) as well. Considering these details, on average, a gauge accounts for 218 km2 (= 30,389/139). Based on this information, you have fitted your 1 km gridded surface. Is this what you want to achieve in this manuscript? If I have misunderstood, do correct me.

8) Table 1 is a good piece of work in this manuscript. It would be more useful for the readers of this manuscript if the authors added a few more sections to outline more about the datasets (e.g., ANUClimate and AGCD) that the authors have listed in Table 1. Moreover, I suggest the authors work with the handling editor to find a better title for Table 1.

9) The spatial and temporal resolutions of ANUClimate are 1 km and 24 h, respectively. What if we disaggregate the daily grid of ANUClimate to an hourly grid using the existing hourly rain gauges? Wouldn't it yield the grid surface of your interest (i.e., 1 km in spatial and 1 h in temporal)? How would this differ from your methodology?

10) Refer to PART IV

---

## Community Comment (CC5)

1-9) Refer to PART I, II, and III

10)  As per the authors, the radar rainfall shows the rainfall in the atmosphere instead of the rainfall reaching the ground. Moreover, as per the authors, the Bureau of Meteorology has found that there are errors in the radar-derived rainfall data, showing unreasonable high rainfall values in some areas. Due to this reason, the authors acknowledge that the hourly rainfall accumulating from radar data was not used in the analysis[LN 146-151].

With the advancement of technology and theoretical development, researchers have been employing ML and AI algorithms to learn from data. Considering this trend, wouldn't ML/AI algorithms improve the development of gridded rainfall surfaces of your interest (i.e., 1 km in spatial and 1 h in temporal)? With the availability of daily rainfall gauges, hourly rainfall gauges, and hourly radar rainfall data, wouldn't it be feasible to develop an ML algorithm to relate the radar data and the existing hourly and daily gauged data to predict the correct values of hourly radar values (i.e., 1 km in spatial and 1 h in temporal)?  How would this differ from your methodology?

11) Refer to PART V

---

## Community Comment (CC6)

1-10) Refer to PART I-IV

11) A few instances of ambiguous citing have been found in the current version of the manuscript. For example,

Statement #1-LN 18-21
High resolution temporal and spatial representations of precipitation data are required in many hydrological applications, such as modelling flood inundation (Jhong et al., 2017; Pappenberger et al., 2005), analysing catchment responses in rainfall-runoff models (Xu et al., 2022; Pappenberger et al., 2005; Acharya et al., 2019), and forecasting extreme events and natural hazards (Ficchi et al., 2016; Mukherjee et al., 2018).

Statement #2-LN 365-366
Hourly rainfall data are essential for many hydrological, ecological, and meteorological applications (Lewis et al., 2018; Hatono et al., 2022).

Why would Lewis et al. (2018) and Hatono et al. (2022) not fit into statement #1 in the current version of the manuscript? I suggest the authors work with the handling editor to resolve these issues.

12) As per the authors, the independent variable transformation for the DEM is x/a, where "x" is the DEM value, and "a" is the transformation parameter. In this study, "a" was set as 10,000 to reduce the impact of the DEM on the hourly rainfall surfaces. The usual value recommended for interpolating monthly and daily data is 1000 (LN 202-203).

The latitude and longitude values are in decimal degrees. The DEM is in liner unit (e.g., m). The rainfall data is in mm. Therefore, there exists a non-normality issue with the data that have been used to fit the surface. I am not sure if the authors are aware of this. Does this lead to setting a random number to the value of "a" (e.g., 9999 instead of 10000)? The current version of the manuscript doesn't justify the magnitude of "a". Why would it jump to 10,000 from 1000 just because the work is hourly instead of daily or monthly?

13) Refer to PART VI

---

## Author Comment (AC1)

**Revision Note 2**

This report addresses the comments from the reviewer on our submission HESS-2024-228, "Very high spatial and temporal resolution rainfall data for accurate flood inundation modelling"

**This is a well-structured and straightforward paper. I have no doubt that the authors' results are likely to be very useful to flood-risk and flood-disaster managers. If I understand correctly, the authors derive a new 15-year high-res spatiotemporal precip. dataset from existing rain gauge and reanalysis data in an Australian location which is prone to short timescale flooding and hence where good hydrological precipitation/flood modelling is highly desirable. The authors compare the resulting product with existing alternatives and find that it is superior when their specific metrics are used. I do not disagree in principle with the authors conclusions, nor do I find fault with the methodology used to produce the CHRain dataset or used to compare the CHRain dataset with BARRA-SY, ANUClimate, and AGCD.**

**My sole reservation is with this study's contribution to the current state-of-the-art. The datasets used are all well-established. The interpolation is done via an off-the-shelf software tool. The temporal downscaling is done "*using the hourly distribution pattern from the nearest hourly station*" and applying some reasonable quality control which is not an innovation.**

**The authors are correct in that: "The proposed method opens an opportunity to develop high resolution spatiotemporal rainfall datasets for other regions" which are essential for developing "detailed flood modelling". However, I find this paper to be more a successful, and very useful, application of an established methodology that a progression beyond the state of the art.**

Thank you for your evaluation.

This paper aims to develop a methodology to generate a 1km hourly rainfall dataset for the Richmond River catchment in Australia, which can be applied to other areas. Since the catchment has a coarse distribution of hourly rain gauges, we have to disaggregate daily data at some daily stations to hourly. Unfortunately, improving the disaggregation method or the quality control process is not within the scope of this paper, but we suggest including better approaches where they are doable.

Even though the study applied the thin-plate spline method, which has been previously used for spatial interpolation of climatic variables, this paper is the first to test the ability of the ANUSPLIN program to generate 1km hourly rainfall surfaces. Moreover, the revised version of the paper has incorporated two key methodological advancements. Firstly, the paper now describes the joint optimisation of the supporting DEM resolution and elevation scaling, now assessed to be 5km and 4,000 respectively. Secondly, the paper has now implemented a robust process to automatically detect false zero data values by analysing the hourly occurrence data. False zeroes are a very common problem with rainfall observations. They are hard to detect by applying simple thresholds. The automated occurrence analysis detected and removed 42,913 false zeroes from 157,378,817 observations (around 0.26%). This led to a significant improvement in the accuracy of the analysis shown in Figure 6.

We edited sub-section **2.6 Generate hourly splines using ANUSPLIN** in the revised manuscript as:

The hourly rainfall splines were generated using ANUSPLIN version 4.4 (Hutchinson and Xu, 2004). There are four main steps to generate daily and hourly splines, including preparing the input data (.dat) files and preparing the command (.cmt) files, running the spline program to generate interpolating parameters and flagging bad zero values, rerunning the spline program with the flag file results from the first fit using spline program, and running the lapgrd program to generate rainfall surfaces. For the hourly rainfall surfaces, we ran the ANUSPLIN program daily (24 splines per day) from 30/01/2007 to 31/12/2022.

The details of the setup are:

1. The independent variables include the longitudes, latitudes, and DEM values of the hourly stations. The dependent variables are the measured rainfall values at the hourly stations.

2. For the spline commands, the numbers of knots were set as 90% of the total number of stations, as read from the input data files. The dependent variable transformation was set as the square root of the data surface to comply with the positive skew of the rainfall values, often including many zeroes, and to ensure that the fitted values are always non-negative Hutchinson et al. (2009).

3. Since the hourly data contain a significantly higher number of zeros and some of the zeros values are artifacts (bad zeros), the flag file resulted from the first fit using the spline program was fed into a second spline fit, where the flagged bad zeros could be removed automatically. There were 42,193 bad zeroes out of 15,737,817 data values in our hourly dataset, amounting to 0.26% of the data values.

4. The optimised parameters from the spline program and the 1 km smoothed DEM were input into the lapgrd program to generate the rainfall grids.

We added sub-section 2.5 in the section 2 Data and methods in the revised manuscript as:

[revised manuscript text omitted]

The analysis on the impact of including the DEM as an independent variable also supports the previous conclusion. Table 5 shows that the optimal trivariate analysis reduced the MAPE by about 2%, during both light, medium, and heavy hourly rainfalls. When the elevation was included in the interpolation, the MAR decreased by 15% and 18% during the light and medium to heavy rainfalls, respectively. The transformation parameter of 4000 and the optimal DEM focal distance of 5 km were used in the ANUSPLIN program to generate the CHRain surfaces for further analysis.

*Table 5. Comparison between bivariate and optimal trivariate analyses on light (0-1 mmh-1) and medium to high rainfalls (>1 mmh-1).*

| | Bivariate | | Trivariate | |
|---|---|---|---|---|
| | MAPE | MAR | MAPE | MAR |
| 0-1 mmh$^{-1}$ | 0.2008 | 0.0548 | 0.1972 | 0.047 |
| > 1 mmh$^{-1}$ | 0.5441 | 0.5394 | 0.5348 | 0.4432 |

We also added in lines 393-404 in the Discussion in the revised manuscript as:

"Compared to daily or monthly data, the hourly data contains significantly more zero values, which can increase the instability of the interpolation model. This paper is the first to test the ability of the ANUSPLIN program to generate hourly rainfall surfaces. It has also incorporated a robust automated process to remove false zeroes from the data. False zeroes are a very common problem with rainfall observations. They are hard to detect by applying simple thresholds. The method proposed in this study has been successfully applied to generate a 1 km hourly gridded rainfall dataset for a larger area. Hourly rainfall data are essential for many hydrological, ecological, and meteorological applications (Lewis et al., 2018; Hatono et al., 2022).

Including elevation data enhances the performance of the thin-spline interpolation model in generating hourly rainfall surfaces, more significantly during larger rainfalls. While the response of the splines to the topography during light rain days is quite broad, the elevation data has greater impacts during larger rain days and results in the clear optimal values for the DEM transformation parameter and the smoothing distance. There are higher resolution DEMs than the 1 km used in the analysis in this paper.

However, the result suggests including finer topographic data does not result in better rainfall surfaces at higher spatial resolution. For our study area, the optimal values for the elevation transformation parameter and the DEM focal distance are 4000 and 5 km, respectively."

**Minor comments and typos:**

**Line    31) "Observation" should be "observations"**

**33) "… more than 20 years" should be "… more than 20 years long"**

**65) "showed to improve" should be "appeared to improve"**

**84) "An accurate high resolution spatial and temporal resolution rainfall" should be "An accurate high spatial and temporal resolution rainfall"**

**104) spurious comma after "especially".**

**132) in "an" area…**

**176) "Disaggregate daily rainfall data to hourly" should be "Disaggregation of daily rainfall to hourly" or something similar…**

**189) "After cleaning, disaggregating, and detailed quality control of the data" should be "After cleaning, disaggregating, and completing a detailed quality control of the data" I think…**

**Figure 7) The first and last row could be removed without loss of clarity…**

We adapted all the minor comments in the revised manuscript.

---

## Author Comment (AC2)

**Revision Note 1**

This report addresses the comments from the reviewer on our submission HESS-2024-228, "Very high spatial and temporal resolution rainfall data for accurate flood inundation modelling"

The manuscript presents an hourly rainfall product at 1 km resolution, derived by disaggregating daily station data (where needed) to an hourly scale and interpolating using a thin-spline method. The authors compare their dataset to existing rainfall products and demonstrate improvements. The manuscript is generally well-written and structured. However, I find the novelty of the study lacking. The thin-plate spline method has been previously used for spatial interpolation of climatic variables (by Hutchinson, one of the authors), and the authors do not propose any methodological advancements (unless I overlooked them) from what has been published in the past. Similarly, the disaggregation approach appears overly simplistic compared to more sophisticated state-of-the-art methods. Given the limited methodological innovation, I question the suitability of this paper for HESS. A more appropriate venue might be a journal focusing on dataset development.

Thank you for your evaluation.

Even though the study applied the thin-plate spline method, which has been previously used for spatial interpolation of climatic variables, this paper is the first to test the ability of the ANUSPLIN program to generate 1km hourly rainfall surfaces. Moreover, the revised version of the paper has incorporated two key methodological advancements. Firstly, the paper now describes the joint optimisation of the supporting DEM resolution and elevation scaling, now assessed to be 5 km and 4,000 respectively. Secondly, the paper has now implemented a robust process to automatically detect false zero data values by analysing the hourly occurrence data. False zeroes are a very common problem with rainfall observations. They are hard to detect by applying simple thresholds. The automated occurrence analysis detected and removed 42,913 false zeroes from 157,378,817 observations (around 0.26%). This led to a significant improvement in the accuracy of the analysis shown in Figure 6 in the old and revised manuscript.

We addressed the comments from the reviewer in detail below.

**1. I suggest revising the title. A 1 km resolution is not necessarily "very high," and the precipitation dataset (why focus only on rainfall?) has applications beyond flood inundation modeling.**

We changed the title of the paper to: "Elevation dependent spatial interpolation of hourly rainfall for accurate flood inundation modelling", reflecting the enhanced focus on the detailed analysis of the topographic dependence.

**2. Lines 9-10: The CHRain dataset is compared to other gridded datasets available in Australia, but does it outperform them? This should be clarified.**

We edited lines 9-10 in the revised manuscript as: "The spatial and temporal analyses indicate that the CHRain dataset outperforms other gridded datasets currently available in Australia in

representing the sub-grid distribution as well as the daily and hourly variation of rainfall across the study area"

3. Line 10: A correlation of 0.948 is quite high, but was it calculated using all the data involved in interpolation and merging? A "leave-one-out" validation approach would provide a more reliable assessment.

In our study area, the hourly data gauges are sparse, so we have to disaggregate some daily data into hourly. The average rain gauge density in the catchment is 143 gauges/30,389 km2 or 1 gauge/212 km2. Therefore, we want to use all the available hourly data to improve the coverage of the rainfall splines. Instead, we evaluated the sum of 24-hour rainfall with daily measurements at 169 gauges that were not used in the interpolation. The revised paper now reports "leave-one-out" validation statistics to supplement these comparisons.

4. Line 20: The same citation is used for two different statements on lines 19 and 20. Please verify.

We removed the citation in line 20 in the revised manuscript.

5. Lines 21-23: This section discusses temporal variability, but what about spatial variability? Given its impact on flood modeling, I recommend considering the following references: <a href="https://doi.org/10.5194/hess-17-2195-2013">https://doi.org/10.5194/hess-17-2195-2013</a> and <a href="https://doi.org/10.5194/hess-21-1559-2017">https://doi.org/10.5194/hess-17-2195-2013</a> and <a href="https://doi.org/10.5194/hess-21-1559-2017">https://doi.org/10.5194/hess-17-2195-2013</a> and

There were not many studies that investigated the impacts of the spatial resolution of rainfall data on hydrological applications, more specifically on the performance of hydrodynamic models. Thank you for your reference suggestion. We added these in the revised manuscript in line 27 as:

"Peleg et al. (2013) analysed the subpixel rain distribution by comparing the data from radar with point measurements at high density gauges. The results shows that a density of 3 rain gauges per radar pixel (4 km2) will allow an adequate presentation of radar rainfall."

"Peleg et al (2017) indicated a valuable contribution of spatial distribution of rainfall (26% contribution) on the total variability of modelled urban drainage network."

**6. Lines 26-27: This sentence seems disconnected from the surrounding discussion.**

We removed lines 26-27 in the revised manuscript.

**7. Lines 29-30: The phrase "1 km to 12 km" is unclear. Since stations provide pointscale data and radar typically operates at 1 km resolution (but may not cover all of Australia), this should be clarified.**

Details of the gridded datasets available in Australia and used in the analysis were mentioned in Table 1 in the old and revised manuscripts. To avoid confusion, we removed the text "1 km to 12 km" in the Introduction of the revised manuscript.

8. Lines 72-75: GCMs are not designed to simulate observed rainfall. The current wording is misleading, and I suggest removing references to GCMs in this context.

We removed line 72-75 in the revised manuscript.

**9. Line 99: "Changes significantly" should be quantified. What is the elevation difference?**

We added the text in the revised manuscript: "The elevation ranges between -6.065 m and 934.6 m across the catchment."

**10. Sections 2.1 and 2.5. Do you simply apply the thin-plate spline as suggested by Hutchinson? If so, what is the novelty of your approach?**

As described above, the revised version of the paper has incorporated two key methodological advancements. It now describes the joint optimisation of the supporting DEM resolution and elevation scaling, now assessed to be 5 km and 4,000 respectively. It has also implemented a new robust process to automatically detect false zero data values by analysing the hourly occurrence data.

We edited sub-**section 2.6 Generate hourly splines using ANUSPLIN** in the revised manuscript as:

The hourly rainfall splines were generated using ANUSPLIN version 4.4 (Hutchinson and Xu, 2004). There are four main steps to generate daily and hourly splines, including preparing the input data (.dat) files and preparing the command (.cmt) files, running the spline program to generate interpolating parameters and flagging bad zero values, rerunning the spline program with the flag file results from the first fit using spline program, and running the lapgrd program to generate rainfall surfaces. For the hourly rainfall surfaces, we ran the ANUSPLIN program daily (24 splines per day) from 30/01/2007 to 31/12/2022.

The details of the setup are:

1. The independent variables include the longitudes, latitudes, and DEM values of the hourly stations. The dependent variables are the measured rainfall values at the hourly stations.

2. For the spline commands, the numbers of knots were set as 90% of the total number of stations, as read from the input data files. The dependent variable transformation was set as the square root of the data surface to comply with the positive skew of the rainfall values, often including many zeroes, and to ensure that the fitted values are always non-negative Hutchinson et al. (2009).

3. Since the hourly data contain a significantly higher number of zeros and some of the zeros values are artifacts (bad zeros), the flag file resulted from the first fit using the spline program was fed into a second spline fit, where the flagged bad zeros could be removed automatically. There were 42,193 bad zeroes out of 15,737,817 data values in our hourly dataset, amounting to 0.26% of the data values.

4. The optimised parameters from the spline program and the 1 km smoothed DEM were input into the lapgrd program to generate the rainfall grids.

We added sub-section 2.5 in the section 2 Data and methods in the revised manuscript as:

[revised manuscript text omitted]

The analysis on the impact of including the DEM as an independent variable also supports the previous conclusion. Table 5 shows that the optimal trivariate analysis reduced the MAPE by about 2%, during both light, medium, and heavy hourly rainfalls. When the elevation was included in the interpolation, the MAR decreased by 15% and 18% during the light and medium to heavy rainfalls, respectively. The transformation parameter of 4000 and the optimal DEM focal distance of 5 km were used in the ANUSPLIN program to generate the CHRain surfaces for further analysis.

Table 5. Comparison between bivariate and optimal trivariate analyses on light (0-1 mmh-1) and medium to high rainfalls (>1 mmh-1).

|                       | Bivar  | iate   | Trivariate |        |  |  |
|-----------------------|--------|--------|------------|--------|--|--|
|                       | MAPE   | MAR    | MAPE       | MAR    |  |  |
| 0-1 mmh⁻¹             | 0.2008 | 0.0548 | 0.1972     | 0.047  |  |  |
| > 1 mmh -1 | 0.5441 | 0.5394 | 0.5348     | 0.4432 |  |  |

We also added in lines 393-404 in the Discussion in the revised manuscript as:

"Compared to daily or monthly data, the hourly data contains significantly more zero values, which can increase the instability of the interpolation model. This paper is the first to test the ability of the ANUSPLIN program to generate hourly rainfall surfaces. It has also incorporated a robust automated process to remove false zeroes from the data. False zeroes are a very common problem with rainfall observations. They are hard to detect by applying simple thresholds. The method proposed in this study has been successfully applied to generate a 1 km hourly gridded rainfall dataset for a larger area. Hourly rainfall data are essential for many hydrological, ecological, and meteorological applications (Lewis et al., 2018; Hatono et al., 2022).

Including elevation data enhances the performance of the thin-spline interpolation model in generating hourly rainfall surfaces, more significantly during larger rainfalls. While the response of the splines to the topography during light rain days is quite broad, the elevation data has greater impacts during larger rain days and results in the clear optimal values for the DEM transformation parameter and the smoothing distance. There are higher resolution DEMs than the 1 km used in the analysis in this paper. However, the result suggests including finer topographic data does not result in better rainfall surfaces at higher spatial resolution. For our study area, the optimal values for the elevation transformation parameter and the DEM focal distance are 4000 and 5 km, respectively."

**11. Section 2.4: If the closest station is 10 km away (just giving an example), the correlation may be too low for reliable disaggregation... A sensitivity analysis using stations at varying distances could provide insights into the method's limitations.**

We agree that the density of hourly gauges is coarse in some areas in our catchment. We have no better option than to use the rainfall pattern from the nearest hourly station to a daily station to disaggregate the daily data at that station. To reduce the uncertainty of choosing the disaggregation, we also used the observed movement of rainfall from the radar data to select suitable nearby hourly gauges to disaggregate data from daily to hourly (mentioned in Section 2.4 in the old and revised manuscripts). Since we don't have many stations to choose from in the areas and the 2nd nearest hourly station can be much further away from the nearest one, it will not be beneficial to do a sensitivity analysis using stations at varying distances to improve the disaggregated data.

**12. Line 203: Why is alpha not treated as a calibration parameter?**

The revised paper now optimises the alpha parameter, as well as the elevation scaling. This is described in point 10.

**13. Lines 231-232: The manuscript reports too many goodness-of-fit measures. Why include both NSE and KGE, for instance? I suggest focusing on two distinct indices that provide complementary information.**

The NSE metric is popularly used in other studies to compare modelled and observed rainfall data (i.e., Hatono et al., 2022). We removed the KGE metric in the revised manuscript to reduce the complication.

---

## Author Comment (AC3)

**Dear Editor,**

We appreciate the effort of Sivarajah Mylevaganam to read and review our submission HESS-2024-228, "Very high spatial and temporal resolution rainfall data for accurate flood inundation modelling".

However, while some comments from Sivarajah Mylevaganam are helpful, others are out of the scope of our study or show a lack of knowledge in the field. For example, improving the resolution of Landsat images is not related to generating higher-resolution rainfall surfaces.

We decided to provide brief explanations for these comments to avoid confusion for SM and other future readers.

**Regarding comments 1, 2, and 3 on the academic writing and presentation style:**

We are research scientists from the Commonwealth Scientific and Industrial Research Organisation (Australia's national science agency) and The Australian National University with solid knowledge of hydrology. We have published many journal papers in Q1 journals, so we are confident in our academic writing style to provide a clear and good-quality manuscript without flaws. All the maps, figures, and tables in the manuscript are created by the authors. We confirmed that Figure 1 was created using ArcMap, with the shapefile showing locations of hourly and daily gauge stations and the catchment domain, and the base map showing the topography of the study area.

**Regarding comments 5 and 6 on the quality of the hourly and daily datasets:**

The hourly and daily rainfall datasets were provided by the Bureau of Meteorology. Because of the artifact of measuring devices, the data can contain unreasonably high or low (including negative) values. Therefore, quality control was applied to the datasets before using them in the interpolation model. The hourly and daily thresholds of rainfall values were set after considering the weather conditions of the catchment, observing plots of hourly time series, and comparing data from hourly gauges with other nearby gauges. These thresholds can vary for different areas and different catchments so they should not be referred anywhere else. Moreover, the revised version of the paper has now implemented a robust process to automatically detect false zero data values by analysing the hourly occurrence data. False zeroes are a very common problem with rainfall observations. They are hard to detect by applying simple thresholds. The automated occurrence analysis detected and removed 42,913 false zeroes from 157,378,817 observations (around 0.26%). This led to a significant improvement in the accuracy of the analysis shown in Figure 6.

**Regarding comments 9 and 10 on the scope of the study:**

The scope of our study is not to focus on disaggregating the daily grid (e.g., ANUClimate) to an hourly grid nor applying AI/ML to improve the interpolation of gridded rainfall surfaces. To clarify this further, we cited other work on the disaggregation of gridded datasets in the introduction of the manuscript:

Acharya, S. C., Nathan, R., Wang, Q. J., and Su, C.-H.: Temporal disaggregation of daily rainfall measurements using regional reanalysis for hydrological applications, Journal of Hydrology, 610, 127 867, https://doi.org/https://doi.org/10.1016/j.jhydrol.2022.127867, 2022.

Westra, S., Mehrotra, R., Sharma, A., and Srikanthan, R.: Continuous rainfall simulation: 1. A regionalized subdaily disaggregation approach, Water Resources Research, 48, https://doi.org/https://doi.org/10.1029/2011WR010489, 2012.

Breinl, K. and Di Baldassarre, G.: Space-time disaggregation of precipitation and temperature across different climates and spatial scales, Journal of Hydrology: Regional Studies, 21, 126–146, https://doi.org/https://doi.org/10.1016/j.ejrh.2018.12.002, 2019.

Moreover, the radar data in the study area have some artifacts as shown in the comparison with other gridded datasets, it is not feasible to develop an AI/ML algorithm to relate the radar data and the existing hourly and daily gauged data to predict the correct values of hourly radar values.

**Regarding other comments including comment 12:**

The revised version of the paper has incorporated two key methodological advancements. Firstly, the paper now describes the joint optimisation of the supporting DEM resolution and elevation scaling, now assessed to be 5km and 4,000 respectively. Secondly, the paper has now implemented a robust process to automatically detect false zero data values by analysing the hourly occurrence data. False zeroes are a very common problem with rainfall observations. They are hard to detect by applying simple thresholds. The automated occurrence analysis detected and removed 42,913 false zeroes from 157,378,817 observations (around 0.26%). This led to a significant improvement in the accuracy of the analysis shown in Figure 6.

Accordingly, we have revised the title in the revised manuscript to: "**Elevation dependent** spatial interpolation of hourly rainfall for accurate flood inundation modelling".

This reflects the enhanced focus on the detailed analysis of topographic dependence conducted by the revised paper. We have also included a detailed analysis of the impacts of elevation transformation on the performance of the interpolation model in the revised manuscript, including calibration of the supporting DEM resolution and calibration of the elevation transformation parameter *a*.

We added sub-section 2.5 in the section 2 Data and methods in the revised manuscript as:

[revised manuscript text omitted]

The analysis on the impact of including the DEM as an independent variable also supports the previous conclusion. Table 5 shows that the optimal trivariate analysis reduced the MAPE by about 2%, during both light, medium, and heavy hourly rainfalls. When the elevation was included in the interpolation, the MAR decreased by 15% and 18% during the light and medium to heavy rainfalls, respectively. The transformation parameter of 4000 and the optimal DEM focal distance of 5 km were used in the ANUSPLIN program to generate the CHRain surfaces for further analysis.

Table 5. Comparison between bivariate and optimal trivariate analyses on light (0-1 mmh-1) and medium to high rainfalls (>1 mmh-1).

|                       | Bivar  | riate  | Trivariate |        |  |  |
|-----------------------|--------|--------|------------|--------|--|--|
|                       | MAPE   | MAR    | MAPE       | MAR    |  |  |
| 0-1 mmh -1 | 0.2008 | 0.0548 | 0.1972     | 0.047  |  |  |
| > 1 mmh -1 | 0.5441 | 0.5394 | 0.5348     | 0.4432 |  |  |

We also added in lines 393-404 in the Discussion in the revised manuscript as:

"Compared to daily or monthly data, the hourly data contains significantly more zero values, which can increase the instability of the interpolation model. This paper is the first to test the ability of the ANUSPLIN program to generate hourly rainfall surfaces. It has also incorporated a robust automated process to remove false zeroes from the data. False zeroes are a very common problem with rainfall observations. They are hard to detect by applying simple thresholds. The method proposed in this study has been successfully applied to generate a 1 km hourly gridded rainfall dataset for a larger area. Hourly rainfall data are essential for many hydrological, ecological, and meteorological applications (Lewis et al., 2018; Hatono et al., 2022).

Including elevation data enhances the performance of the thin-spline interpolation model in generating hourly rainfall surfaces, more significantly during larger rainfalls. While the response of the splines to the topography during light rain days is quite broad, the elevation data has greater impacts during larger rain days and results in the clear optimal values for the DEM transformation parameter and the smoothing distance. There are higher resolution DEMs than the 1 km used in the analysis in this paper. However, the result suggests including finer topographic data does not result in better rainfall surfaces at higher spatial resolution. For our study area, the optimal values for the elevation transformation parameter and the DEM focal distance are 4000 and 5 km, respectively."

---

## Author Response (AR1)

**Revision Note 1**

This report addresses the comments from the reviewer 1 on our submission HESS-2024-228, "Very high spatial and temporal resolution rainfall data for accurate flood inundation modelling"

**The manuscript presents an hourly rainfall product at 1 km resolution, derived by disaggregating daily station data (where needed) to an hourly scale and interpolating using a thin-spline method. The authors compare their dataset to existing rainfall products and demonstrate improvements. The manuscript is generally well-written and structured. However, I find the novelty of the study lacking. The thin-plate spline method has been previously used for spatial interpolation of climatic variables (by Hutchinson, one of the authors), and the authors do not propose any methodological advancements (unless I overlooked them) from what has been published in the past. Similarly, the disaggregation approach appears overly simplistic compared to more sophisticated state-of-the-art methods. Given the limited methodological innovation, I question the suitability of this paper for HESS. A more appropriate venue might be a journal focusing on dataset development.**

Thank you for your evaluation.

Even though the study applied the thin-plate spline method, which has been previously used for spatial interpolation of climatic variables, this paper is the first to test the ability of the ANUSPLIN program to generate 1km hourly rainfall surfaces. Moreover, the revised version of the paper has incorporated two key methodological advancements. Firstly, the paper now describes the joint optimisation of the supporting DEM resolution and elevation scaling, now assessed to be 5 km and 4,000 respectively. Secondly, the paper has now implemented a robust process to automatically detect false zero data values by analysing the hourly occurrence data. False zeroes are a very common problem with rainfall observations. They are hard to detect by applying simple thresholds. The automated occurrence analysis detected and removed 42,913 false zeroes from 157,378,817 observations (around 0.26%). This led to a significant improvement in the accuracy of the analysis shown in Figure 6 in the old and revised manuscript.

We addressed the comments from the reviewer in detail below.

1. **I suggest revising the title. A 1 km resolution is not necessarily "very high," and the precipitation dataset (why focus only on rainfall?) has applications beyond flood inundation modeling.**

We changed the title of the paper to: "**Elevation dependent spatial interpolation of hourly rainfall for accurate flood inundation modelling**", reflecting the enhanced focus on the detailed analysis of the topographic dependence.

2. **Lines 9-10: The CHRain dataset is compared to other gridded datasets available in Australia, but does it outperform them? This should be clarified.**

We edited lines 9-10 in the old manuscript (lines 14-16 in the revised manuscript) as: "The spatial and temporal analyses indicate that the CHRain dataset outperforms other gridded datasets

currently available in Australia in representing the sub-grid distribution as well as the daily and hourly variation of rainfall across the study area"

**3. Line 10: A correlation of 0.948 is quite high, but was it calculated using all the data involved in interpolation and merging? A "leave-one-out" validation approach would provide a more reliable assessment.**

In our study area, the hourly data gauges are sparse, so we have to disaggregate some daily data into hourly. The average rain gauge density in the catchment is 143 gauges/30,389 $km^2$ or 1 gauge/212 $km^2$. Therefore, we want to use all the available hourly data to improve the coverage of the rainfall splines. Instead, we evaluated the sum of 24-hour rainfall with daily measurements at 169 gauges that were not used in the interpolation. The revised paper now reports "leave-one-out" validation statistics in the ANUSPLIN program to supplement these comparisons.

**4. Line 20: The same citation is used for two different statements on lines 19 and 20. Please verify.**

We removed the citation in line 20 in the revised manuscript.

**5. Lines 21-23: This section discusses temporal variability, but what about spatial variability? Given its impact on flood modeling, I recommend considering the following references: https://doi.org/10.5194/hess-17-2195-2013 and https://doi.org/10.5194/hess-21-1559-2017**

There were not many studies that investigated the impacts of the spatial resolution of rainfall data on hydrological applications, more specifically on the performance of hydrodynamic models. Thank you for your reference suggestion. We added these in the revised manuscript in lines 28-33 as:

"Peleg et al. (2013) analysed the subpixel rain distribution by comparing the data from radar with point measurements at high density gauges. The results shows that a density of 3 rain gauges per radar pixel (4 km x 4 km) will allow an adequate presentation of radar rainfall."

"Peleg et al (2017) indicated a valuable contribution of spatial distribution of rainfall (26% contribution) on the total variability of modelled urban drainage network."

**6. Lines 26-27: This sentence seems disconnected from the surrounding discussion.**

We removed lines 26-27 in the revised manuscript.

**7. Lines 29-30: The phrase "1 km to 12 km" is unclear. Since stations provide point-scale data and radar typically operates at 1 km resolution (but may not cover all of Australia), this should be clarified.**

Details of the gridded datasets available in Australia and used in the analysis were mentioned in Table 1 in the old and revised manuscripts. To avoid confusion, we removed the text "1 km to 12 km" in the Introduction of the revised manuscript.

**8. Lines 72-75: GCMs are not designed to simulate observed rainfall. The current wording is misleading, and I suggest removing references to GCMs in this context.**

We removed line 72-75 in the revised manuscript.

9. **Line 99: "Changes significantly" should be quantified. What is the elevation difference?**

We added the text in the revised manuscript lines 101-102 as: "The elevation ranges between 0 m and 934.6 m across the catchment."

10. **Sections 2.1 and 2.5. Do you simply apply the thin-plate spline as suggested by Hutchinson? If so, what is the novelty of your approach?**

As described above, the revised version of the paper has incorporated two key methodological advancements. It now describes the joint optimisation of the supporting DEM resolution and elevation scaling, now assessed to be around 5 km and 4,000 respectively. It has also implemented a new robust process to automatically detect false zero data values by analysing the hourly occurrence data.

We edited **Sub-section 2.3 Quality control for the hourly rainfall data** in the revised manuscript lines 174-188 as:

[revised manuscript text omitted]

We added **Sub-section 3.2** in **Section 3 Results** in the revised manuscript lines 294-310 as:

**3.2 Impacts of topography on the spatial interpolation of hourly rainfall splines**
Table 3 and Table 4 show the Square RooT of the average Generalised Cross Validation (RTGCV) of the trivariate spline model for light rainfall days and medium to high rainfall days as a function of DEM focal distance and elevation scaling, as derived in the initial analyses with no removal of false zeroes. The light rainfall days indicate a very broad dependence on the topographic parameters with an optimum DEM focal distance around 10 km or possibly larger. On the other hand, the medium to high rainfall days indicate an optimum DEM focal distance of around 5 km and an optimum elevation scaling of around 4000. This suggests that topography plays an important role in interpolating larger rainfalls while the response of smaller rainfalls to topography is fairly flat. The daily average 1 $mmh^{-1}$ threshold appears to be an effective discriminator of light and medium to high rainfall days. Setting a lower threshold gave rise to multiple local minima in the RTGCV patterns for days with average hourly rainfall greater than 0.5 $mmh^{-1}$. These tables were recalculated after false zeroes were removed by the spline occurrence analysis described above, with DEM focal distance set to 5 km and elevation scaling set to 4000. The resulting patterns were similar to those shown in Table 3 and Table 4, with an optimum

DEM focal distance of around 5 km and a slightly larger elevation scaling of around 5000. There was little difference between the performance with these two elevation scales. All the remaining analyses were completed on the data with false zeroes removed, using the initially determined 5 km DEM focal distance and elevation scaling of 4000.

The impact of including the DEM as an independent variable was further quantified in Table 5. It shows that, compared to the bivariate analysis, the optimal trivariate analysis reduced the MAPE by about 4% for light rainfall days and by about 2% for medium to heavy rainfall days. The trivariate analysis reduced the MAR by about 16% across all days.

Table 3. Performance of the interpolation model with different elevation transformation parameters and elevation smoothing scales for light rain days (0-1 mmh$^{-1}$). The minimum values of the RTGCV are shown in bold.

| a | 1 km | 2 km | 3 km | 4 km | 5 km | 6 km | 7 km | 8 km | 9 km | 10 km |
|---|---|---|---|---|---|---|---|---|---|---|
| 1000 | 0.2003 | 0.2005 | 0.1993 | 0.1984 | 0.1981 | 0.1980 | 0.1978 | 0.1978 | 0.1976 | 0.1978 |
| 2000 | 0.1983 | 0.1978 | 0.1981 | 0.1976 | 0.1976 | 0.1973 | 0.1970 | 0.1969 | 0.1969 | **0.1968** |
| 3000 | 0.1975 | 0.1978 | 0.1976 | 0.1974 | 0.1973 | 0.1973 | 0.1973 | 0.1972 | 0.1971 | 0.1967 |
| 4000 | 0.1975 | 0.1976 | 0.1975 | 0.1973 | 0.1972 | 0.1974 | 0.1973 | 0.1970 | 0.1971 | 0.1971 |
| 5000 | 0.1976 | 0.1974 | 0.1973 | 0.1972 | 0.1972 | 0.1972 | 0.1971 | 0.1970 | 0.1971 | **0.1969** |
| 6000 | 0.1975 | 0.1973 | 0.1973 | 0.1972 | 0.1972 | 0.1972 | 0.1970 | 0.1970 | **0.1969** | **0.1969** |
| 7000 | 0.1975 | 0.1973 | 0.1973 | 0.1972 | 0.1972 | 0.1971 | 0.1970 | 0.1970 | **0.1969** | **0.1969** |
| 8000 | 0.1974 | 0.1973 | 0.1972 | 0.1973 | 0.1972 | 0.1971 | 0.1970 | 0.1970 | **0.1969** | 0.1970 |
| 9000 | 0.1975 | 0.1972 | 0.1974 | 0.1972 | 0.1972 | 0.1971 | 0.1970 | 0.1970 | **0.1969** | 0.1970 |
| 10,000 | 0.1974 | 0.1972 | 0.1973 | 0.1972 | 0.1972 | 0.1971 | 0.1970 | 0.1970 | **0.1969** | 0.1970 |

Table 4. Performance of the interpolation model with different elevation transformation parameters and elevation smoothing scales for medium to high rain days (> 1 mmh$^{-1}$). The minimum value of the RTGCV is shown in bold.

| a | 1 km | 2 km | 3 km | 4 km | 5 km | 6 km | 7 km | 8 km | 9 km | 10 km |
|---|---|---|---|---|---|---|---|---|---|---|
| 1000 | 0.5536 | 0.5518 | 0.5485 | 0.5449 | 0.5427 | 0.5438 | 0.5431 | 0.5436 | 0.5423 | 0.5442 |
| 2000 | 0.5429 | 0.5408 | 0.5411 | 0.5385 | 0.5372 | 0.5362 | 0.5374 | 0.5374 | 0.5366 | 0.5403 |
| 3000 | 0.5387 | 0.5393 | 0.5377 | 0.5364 | 0.5359 | 0.5352 | 0.5370 | 0.5370 | 0.5376 | 0.5366 |
| 4000 | 0.5387 | 0.5372 | 0.5366 | 0.5357 | **0.5348** | 0.5359 | 0.5363 | 0.5361 | 0.5369 | 0.5362 |
| 5000 | 0.5369 | 0.5366 | 0.5362 | 0.5351 | 0.5356 | 0.5357 | 0.5362 | 0.5464 | 0.5367 | 0.5359 |
| 6000 | 0.5368 | 0.5356 | 0.5351 | 0.5349 | 0.5359 | 0.5364 | 0.5363 | 0.5465 | 0.5363 | 0.5361 |
| 7000 | 0.5358 | 0.5355 | 0.5351 | 0.5350 | 0.5359 | 0.5364 | 0.5363 | 0.5366 | 0.5362 | 0.5360 |
| 8000 | 0.5356 | 0.5354 | 0.5352 | 0.5354 | 0.5362 | 0.5363 | 0.5363 | 0.5366 | 0.5363 | 0.5360 |
| 9000 | 0.5354 | 0.5354 | 0.5356 | 0.5364 | 0.5367 | 0.5464 | 0.5363 | 0.5365 | 0.5358 | 0.5359 |
| 10,000 | 0.5354 | 0.5353 | 0.5361 | 0.5364 | 0.5365 | 0.5461 | 0.5366 | 0.5366 | 0.5359 | 0.5359 |

Table 5. Comparison between bivariate and optimal trivariate analyses on light (0-1 mmh$^{-1}$) and medium to high rainfalls (>1 mmh$^{-1}$).

| | Bivariate | | Trivariate | |
|---|---|---|---|---|
| | MAPE | MAR | MAPE | MAR |
| 0-1 mmh$^{-1}$ | 0.0884 | 0.0505 | 0.0851 | 0.0420 |
| > 1 mmh$^{-1}$ | 0.9007 | 0.4378 | 0.8816 | 0.3681 |

We also added in lines 413-434 in the **Discussion** in the revised manuscript as:

> Compared to daily or monthly data, the hourly data contains significantly more zeros, which can increase the instability of the interpolation model. This paper is the first to test the ability of the ANUSPLIN program to generate hourly rainfall surfaces. It has also incorporated a robust automated process to remove false zeros from the data. False zeros are a very common problem with rainfall observations. They are hard to detect by applying simple thresholds. The method proposed in this study has been successfully applied to generate a 1 km hourly gridded rainfall dataset for a larger area. Hourly rainfall data are essential for many hydrological, ecological, and meteorological applications (Lewis et al., 2018; Hatono et al., 2022).

> Including elevation data enhances the performance of the thin-spline interpolation model in generating hourly rainfall surfaces, more significantly during larger rainfalls. While the response of the splines to the topography during light rain days is quite broad, the elevation data has greater impacts during larger rain days and results in the clear optimal values for the DEM transformation parameter and the smoothing distance. There are higher resolution DEMs than the 1 km used in the analysis in this paper. However, the result suggests including finer topographic data does not result in better rainfall surfaces at higher spatial resolution. For our study area, the optimal values for the transformation parameter $a$ and the DEM focal distance are around 4000 to 5000 and 5 km, respectively. The optimal DEM focal distance of 5 km is in agreement with the analysis of Sharples et al. (2005), who showed that similarly averaged DEMs with focal distances from 5 to 10 km performed best in interpolating monthly rainfall across Australia. On the other hand, the optimal elevation scaling of around 4000 to 5000 corresponds to a vertical exaggeration of around 20. This is somewhat less than the vertical exaggeration of around 100 found with spatial analyses of rainfall at broader time scales by Hutchinson (1995) and Johnson et al. (2016). This suggests that hourly rainfall, though significantly influenced by elevation, has a less consistent dependence on elevation than rainfall values recorded at broader time scales.

> The initial hourly rainfall occurrence analysis appears to have been effective in detecting and removing the many false zeroes that can arise with automatically recorded hourly rainfall data. This was aided by the limited spatial extent of this rainfall analysis. The detections would likely to be less reliable when applied to sites with no relatively near neighbours.

**11. Section 2.4: If the closest station is 10 km away (just giving an example), the correlation may be too low for reliable disaggregation... A sensitivity analysis using stations at varying distances could provide insights into the method's limitations.**

We agree that the density of hourly gauges is coarse in some areas in our catchment. We have no better option than to use the rainfall pattern from the nearest hourly station to a daily station to disaggregate the daily data at that station. To reduce the uncertainty of choosing the disaggregation, we also used the observed movement of rainfall from the radar data to select suitable nearby hourly gauges to disaggregate data from daily to hourly (mentioned in Section 2.4 in the old and revised manuscripts). Since we don't have many stations to choose from in the areas and the 2nd nearest hourly station can be much further away from the nearest one, it will

not be beneficial to do a sensitivity analysis using stations at varying distances to improve the disaggregated data.

**12. Line 203: Why is alpha not treated as a calibration parameter?**

The revised paper now optimises the alpha parameter, as well as the elevation scaling. This is described in point 10.

**13. Lines 231-232: The manuscript reports too many goodness-of-fit measures. Why include both NSE and KGE, for instance? I suggest focusing on two distinct indices that provide complementary information.**

The NSE metric is popularly used in other studies to compare modelled and observed rainfall data (i.e., Hatono et al., 2022). We removed the KGE metric in the revised manuscript to reduce the complication.

**Revision Note 2**

This report addresses the comments from the reviewer 2 on our submission HESS-2024-228, "Very high spatial and temporal resolution rainfall data for accurate flood inundation modelling"

**This is a well-structured and straightforward paper. I have no doubt that the authors' results are likely to be very useful to flood-risk and flood-disaster managers. If I understand correctly, the authors derive a new 15-year high-res spatiotemporal precip. dataset from existing rain gauge and reanalysis data in an Australian location which is prone to short timescale flooding and hence where good hydrological precipitation/flood modelling is highly desirable. The authors compare the resulting product with existing alternatives and find that it is superior when their specific metrics are used. I do not disagree in principle with the authors conclusions, nor do I find fault with the methodology used to produce the CHRain dataset or used to compare the CHRain dataset with BARRA-SY, ANUClimate, and AGCD.**

**My sole reservation is with this study's contribution to the current state-of-the-art. The datasets used are all well-established. The interpolation is done via an off-the-shelf software tool. The temporal downscaling is done "*using the hourly distribution pattern from the nearest hourly station*" and applying some reasonable quality control which is not an innovation.**

**The authors are correct in that: "The proposed method opens an opportunity to develop high resolution spatiotemporal rainfall datasets for other regions" which are essential for developing "detailed flood modelling". However, I find this paper to be more a successful, and very useful, application of an established methodology that a progression beyond the state of the art.**

Thank you for your evaluation.

This paper aims to develop a methodology to generate a 1km hourly rainfall dataset for the Richmond River catchment in Australia, which can be applied to other areas. Since the catchment has a coarse distribution of hourly rain gauges, we have to disaggregate daily data at some daily stations to hourly. Unfortunately, improving the disaggregation method or the quality control process is not within the scope of this paper, but we suggest including better approaches where they are doable.

Even though the study applied the thin-plate spline method, which has been previously used for spatial interpolation of climatic variables, this paper is the first to test the ability of the ANUSPLIN program to generate 1km hourly rainfall surfaces. Moreover, the revised version of the paper has incorporated two key methodological advancements. Firstly, the paper now describes the joint optimisation of the supporting DEM resolution and elevation scaling, now assessed to be 5 km and 4,000 respectively. Secondly, the paper has now implemented a robust process to automatically detect false zero data values by analysing the hourly occurrence data. False zeroes are a very common problem with rainfall observations. They are hard to detect by applying simple thresholds. The automated occurrence analysis detected and removed 42,913 false zeroes from 157,378,817 observations (around 0.26%). This led to a significant improvement in the accuracy of the analysis shown in Figure 6.

We edited **Sub-section 2.3 Quality control for the hourly rainfall data** in the revised manuscript lines 174-188 as:

[revised manuscript text omitted]

We added **Sub-section 3.2** in **Section 3 Results** in the revised manuscript lines 294-310 as:

**3.2 Impacts of topography on the spatial interpolation of hourly rainfall splines**
Table 3 and Table 4 show the Square RooT of the average Generalised Cross Validation (RTGCV) of the trivariate spline model for light rainfall days and medium to high rainfall days as a function of DEM focal distance and elevation scaling, as derived in the initial analyses with no removal of false zeroes. The light rainfall days indicate a very broad dependence on the topographic parameters with an optimum DEM focal distance around 10 km or possibly larger. On the other hand, the medium to high rainfall days indicate an optimum DEM focal distance of around 5 km and an optimum elevation scaling of around 4000. This suggests that topography plays an important role in interpolating larger rainfalls while the response of smaller rainfalls to topography is fairly flat. The daily average 1 mmh$^{-1}$ threshold appears to be an effective discriminator of light and medium to high rainfall days. Setting a lower threshold gave rise to multiple local minima in the RTGCV patterns for days with average hourly rainfall greater than 0.5 mmh$^{-1}$. These tables were recalculated after false zeroes were removed by the spline occurrence analysis described above, with DEM focal distance set to 5 km and elevation scaling set to 4000. The resulting patterns were similar to those shown in Table 3 and Table 4, with an optimum DEM focal distance of around 5 km and a slightly larger elevation scaling of around 5000. There was little difference between the performance with these two elevation scales. All the remaining analyses were completed on the data with false zeroes removed, using the initially determined 5 km DEM focal distance and elevation scaling of 4000.

The impact of including the DEM as an independent variable was further quantified in Table 5. It shows that, compared to the bivariate analysis, the optimal trivariate analysis reduced the MAPE by about 4% for light rainfall days and by about 2% for medium to heavy rainfall days. The trivariate analysis reduced the MAR by about 16% across all days.

Table 3. Performance of the interpolation model with different elevation transformation parameters and elevation smoothing scales for light rain days (0-1 mmh$^{-1}$). The minimum values of the RTGCV are shown in bold.

| a | 1 km | 2 km | 3 km | 4 km | 5 km | 6 km | 7 km | 8 km | 9 km | 10 km |
|---|------|------|------|------|------|------|------|------|------|-------|
| 1000 | 0.2003 | 0.2005 | 0.1993 | 0.1984 | 0.1981 | 0.1980 | 0.1978 | 0.1978 | 0.1976 | 0.1978 |
| 2000 | 0.1983 | 0.1978 | 0.1981 | 0.1976 | 0.1976 | 0.1973 | 0.1970 | 0.1969 | 0.1969 | **0.1968** |
| 3000 | 0.1975 | 0.1978 | 0.1976 | 0.1974 | 0.1973 | 0.1973 | 0.1973 | 0.1972 | 0.1971 | 0.1967 |
| 4000 | 0.1975 | 0.1976 | 0.1975 | 0.1973 | 0.1972 | 0.1974 | 0.1973 | 0.1970 | 0.1971 | 0.1971 |
| 5000 | 0.1976 | 0.1974 | 0.1973 | 0.1972 | 0.1972 | 0.1972 | 0.1971 | 0.1970 | 0.1971 | **0.1969** |
| 6000 | 0.1975 | 0.1973 | 0.1973 | 0.1972 | 0.1972 | 0.1972 | 0.1970 | 0.1970 | **0.1969** | **0.1969** |
| 7000 | 0.1975 | 0.1973 | 0.1973 | 0.1972 | 0.1972 | 0.1971 | 0.1970 | 0.1970 | **0.1969** | **0.1969** |
| 8000 | 0.1974 | 0.1973 | 0.1972 | 0.1973 | 0.1972 | 0.1971 | 0.1970 | 0.1970 | **0.1969** | 0.1970 |
| 9000 | 0.1975 | 0.1972 | 0.1974 | 0.1972 | 0.1972 | 0.1971 | 0.1970 | 0.1970 | **0.1969** | 0.1970 |
| 10,000 | 0.1974 | 0.1972 | 0.1973 | 0.1972 | 0.1972 | 0.1971 | 0.1970 | 0.1970 | **0.1969** | 0.1970 |

Table 4. Performance of the interpolation model with different elevation transformation parameters and elevation smoothing scales for medium to high rain days (> 1 mmh$^{-1}$). The minimum value of the RTGCV is shown in bold.

| a | 1 km | 2 km | 3 km | 4 km | 5 km | 6 km | 7 km | 8 km | 9 km | 10 km |
|---|------|------|------|------|------|------|------|------|------|-------|
| 1000 | 0.5536 | 0.5518 | 0.5485 | 0.5449 | 0.5427 | 0.5438 | 0.5431 | 0.5436 | 0.5423 | 0.5442 |
| 2000 | 0.5429 | 0.5408 | 0.5411 | 0.5385 | 0.5372 | 0.5362 | 0.5374 | 0.5374 | 0.5366 | 0.5403 |
| 3000 | 0.5387 | 0.5393 | 0.5377 | 0.5364 | 0.5359 | 0.5352 | 0.5370 | 0.5370 | 0.5376 | 0.5366 |
| 4000 | 0.5387 | 0.5372 | 0.5366 | 0.5357 | **0.5348** | 0.5359 | 0.5363 | 0.5361 | 0.5369 | 0.5362 |
| 5000 | 0.5369 | 0.5366 | 0.5362 | 0.5351 | 0.5356 | 0.5357 | 0.5362 | 0.5464 | 0.5367 | 0.5359 |
| 6000 | 0.5368 | 0.5356 | 0.5351 | 0.5349 | 0.5359 | 0.5364 | 0.5363 | 0.5465 | 0.5363 | 0.5361 |
| 7000 | 0.5358 | 0.5355 | 0.5351 | 0.5350 | 0.5359 | 0.5364 | 0.5363 | 0.5366 | 0.5362 | 0.5360 |
| 8000 | 0.5356 | 0.5354 | 0.5352 | 0.5354 | 0.5362 | 0.5363 | 0.5363 | 0.5366 | 0.5363 | 0.5360 |
| 9000 | 0.5354 | 0.5354 | 0.5356 | 0.5364 | 0.5367 | 0.5464 | 0.5363 | 0.5365 | 0.5358 | 0.5359 |
| 10,000 | 0.5354 | 0.5353 | 0.5361 | 0.5364 | 0.5365 | 0.5461 | 0.5366 | 0.5366 | 0.5359 | 0.5359 |

Table 5. Comparison between bivariate and optimal trivariate analyses on light (0-1 mmh$^{-1}$) and medium to high rainfalls (>1 mmh$^{-1}$).

| | Bivariate | | Trivariate | |
|---|-----------|------|------------|------|
| | MAPE | MAR | MAPE | MAR |
| 0-1 mmh$^{-1}$ | 0.0884 | 0.0505 | 0.0851 | 0.0420 |
| > 1 mmh$^{-1}$ | 0.9007 | 0.4378 | 0.8816 | 0.3681 |

We also added in lines 413-434 in the **Discussion** in the revised manuscript as:

Compared to daily or monthly data, the hourly data contains significantly more zeros, which can increase the instability of the interpolation model. This paper is the first to test the ability of the ANUSPLIN program to generate hourly rainfall surfaces. It has also incorporated a robust automated process to remove false zeros from the data. False zeros are a very common problem with rainfall observations. They are hard to detect by applying simple thresholds. The method proposed in this study has been successfully applied to generate a 1 km hourly gridded rainfall dataset for a larger area. Hourly rainfall data are essential for many hydrological, ecological, and meteorological applications (Lewis et al., 2018; Hatono et al., 2022).

Including elevation data enhances the performance of the thin-spline interpolation model in generating hourly rainfall surfaces, more significantly during larger rainfalls. While the response of the splines to the topography during light rain days is quite broad, the elevation data has greater impacts during larger rain days and results in the clear optimal values for the DEM transformation parameter and the smoothing distance. There are higher resolution DEMs than the 1 km used in the analysis in this paper. However, the result suggests including finer topographic data does not result in better rainfall surfaces at higher spatial resolution. For our study area, the optimal values for the transformation parameter $a$ and the DEM focal distance are around 4000 to 5000 and 5 km, respectively. The optimal DEM focal distance of 5 km is in agreement with the analysis of Sharples et al. (2005), who showed that similarly averaged DEMs with focal distances from 5 to 10 km performed best in interpolating monthly rainfall across Australia. On the other hand, the optimal elevation scaling of around 4000 to 5000 corresponds to a vertical exaggeration of around 20. This is somewhat less than the vertical exaggeration of around 100 found with spatial analyses of rainfall at broader time scales by Hutchinson (1995) and Johnson et al. (2016). This suggests that hourly rainfall, though significantly influenced by elevation, has a less consistent dependence on elevation than rainfall values recorded at broader time scales.

The initial hourly rainfall occurrence analysis appears to have been effective in detecting and removing the many false zeroes that can arise with automatically recorded hourly rainfall data. This was aided by the limited spatial extent of this rainfall analysis. The detections would likely to be less reliable when applied to sites with no relatively near neighbours.

**Minor comments and typos:**

**Line    31) "Observation" should be "observations"**

**33) "... more than 20 years" should be "... more than 20 years long"**

**65) "showed to improve" should be "appeared to improve"**

**84) "An accurate high resolution spatial and temporal resolution rainfall" should be "An accurate high spatial and temporal resolution rainfall"**

**104) spurious comma after "especially".**

**132) in "an" area...**

**176) "Disaggregate daily rainfall data to hourly" should be "Disaggregation of daily rainfall to hourly" or something similar...**

**189) "After cleaning, disaggregating, and detailed quality control of the data" should be "After cleaning, disaggregating, and completing a detailed quality control of the data" I think...**

**Figure 7) The first and last row could be removed without loss of clarity...**

We adapted all the minor comments in the revised manuscript.

---

## Author Response (AR2)

**Revision notes**

**The revision is much improved. The added calibration of DEM smoothing scale and elevation scaling, plus the automated false-zero occurrence analysis, strengthen the methodological contribution and the dataset's reliability.**

**The CHRain product shows strong correspondence with observations (e.g., r≈0.949 for hourly comparisons) and clear advantages over BARRA-SY and radar during the 2017 event.**

Thank you for your evaluation. We addressed the comments from the reviewer in detail below.

**Major comments:**
**1. Reproducibility & reusability (scripts/data):**
**Given the practical value of the pipeline (hourly disaggregation → quality control/false-zero removal → ANUSPLIN interpolation → evaluation), please add a Code & Data Availability section and share:**
**o the disaggregation scripts implementing your three criteria and 9am–8am alignment,**
**o the occurrence-analysis code (including the 0.5 threshold and σ=0.25 setting),**
**o ANUSPLIN .cmt files and any DEM smoothing scripts (e.g., focal mean windows), and**
**o a small worked example (one storm day) so users can reproduce Figure/Tables for their area.**
**These elements are all described textually but are not yet available as artifacts; making them public would maximise impact and reuse.**

We provided Python scripts with detailed comments to generate rainfall splines using the ANUSPLIN program, including the processes:

- Disaggregating daily data to hourly data from 9 am to 8 am the next day, and applying the quality control to the combined data (Disagg_d2h.py)
- Generating input files (.dat) from the hourly rainfall dataset to run the ANUSPLIN (Prepare_hourly_input.py)
- Generating the ANUSPLIN command file (.cmd) associated with the input files
- A small sample of one day of rainfall data (24 hours) at 2 rain gauge stations, and the DEM data, so that the user can test the provided scripts
- The 1 km DEM in the analysis was smoothed using the focal mean method in ArcGIS (https://desktop.arcgis.com/en/arcmap/latest/tools/spatial-analyst-toolbox/how-focal-statistics-works.htm)

Users need to contact the Fenner School of Environment & Society (https://fennerschool.anu.edu.au/research/products/anusplin-version-4-4) to get the ANUSPLIN program and the program to run the false zeros analysis and removal. We cannot share the source code of the ANUSPLIN program due to licensing constraints.

We included Python scripts and a sample as suggested by the reviewer in the Code and Data Availability in the revised manuscript (https://doi.org/10.5281/zenodo.17686121).

**2. False-zero detection—report a small audit:**
**Great addition. You report ~0.26% removed; consider adding (i) a short manual audit of a random subset to estimate precision/recall, and (ii) a sensitivity note on the 0.5 threshold. Including a simple confusion matrix for the 30 March 2017 case (Appendix B) would be ideal.**

The false zero removal is a two-step process, including an initial ANUSPLIN occurrence analysis followed by a standalone FORTRAN program (occflg1) that converts the data and fitted occurrence values into standard bad data flags for input into the main ANUSPLIN interpolation. Users need to contact the Fenner School of Environment & Society (https://fennerschool.anu.edu.au/research/products/anusplin-version-4-4) to get the ANUSPLIN program and the program to run the false zeros analysis and removal.

We added the analysis of occurrence-based corrections with hourly radar rainfall data provided by BoM in Section 2.3 in the revised manuscript. Since there are artifacts in the radar rainfall data on 30/3/2017, we apply the analysis on 8 successive days during the two flood events in 2022.

We also conducted a sensitivity analysis on the threshold from 0.4 to 0.8, and they gave similar results to the chosen value of 0.5.

The detailed analysis was added in lines 188-204 and in Table 2:

"The reliability of the occurrence-based corrections was assessed by comparing the analyses with hourly radar rainfall data over eight successive days during the two flood events in 2022 (with the peak around 28/02/2022 and 29/03/2022). Summary statistics are presented in Table \ref{table2}. As noted above, the radar rainfall is not always reliable. The percent occurrence agreement of the raw hourly rainfall data with the radar data ranged between 56% and 72% for six of the eight days, while there were strong occurrence agreements of 92% and 81% on the two high rainfall days on 27/02/2022 and 28/03/2022. This indicates there were major deficiencies in the radar data, except on the heavy rainfall days when significant rainfall was widespread over the data network. Comparing the occurrence corrections with the radar occurrence data showed strong agreement with the original data occurrence agreements, ranging from 43% to 92% on six of the days and 98% and 83% on 27/02/2022 and 28/03/2022. If the corrections were all correct, comparison with the radar data could be expected to assess them as having an accuracy similar to the initial overall agreements between the rainfall data and the radar data. The strong agreement between column 3 and column 5 in Table 2 is consistent with the corrections being in fact highly reliable, with a true accuracy up to around 98% on all eight days. The true reliability maybe somewhat lower on days with less widespread rainfall and less spatially coherent rainfall occurrence patterns. The occurrence corrections were sufficient to improve the overall occurrence agreement with the radar data on the fourth day in the first event and on all three days in the second event. The overall agreement was unchanged for the other days. A range of occurrence thresholds was tested by assessing the overall occurrence agreement of the corrected data with respect to the occurrence threshold. A range of thresholds from 0.4 to 0.8 gave similar results to the chosen value of 0.5. Further refinements are limited by the overall unreliability of the radar rainfall data."

**Table 2.** Summary statistics of occurrence corrections with respect to hourly radar data over five successive days, during the flood event in 2022. Occurrence agreement is calculated as the percentage of the number of agreements in occurrence (both zero or both non-zero) divided by the total number of hourly data values.

| Date | Average Rain (mm/h) | Percent occurrence agreement of raw data with radar data | Number of estimated false zeros | Percent agreement of corrections with radar data | Percent occurrence agreement of corrected data with radar data |
|---|---|---|---|---|---|
| 24/02/2022 | 3.3 | 57 | 75 | 57 | 57 |
| 25/02/2022 | 0.8 | 57 | 36 | 53 | 57 |
| 26/02/2022 | 2.3 | 72 | 54 | 87 | 72 |
| 27/02/2022 | 8.6 | 92 | 102 | 98 | 96 |
| 28/02/2022 | 10.5 | 56 | 85 | 46 | 56 |
| 28/03/2022 | 4.2 | 81 | 90 | 83 | 83 |
| 29/03/2022 | 4.2 | 65 | 94 | 92 | 67 |
| 30/02/2022 | 3.0 | 57 | 83 | 43 | 58 |

**3. Clarify what "outperforms" means in spatial analysis:**
**Where you state "outperforms ANUClimate/AGCD," specify the exact metrics and thresholds (e.g., Bias, Hit Rate, CSI for ≥95th/≥99th percentiles). A compact figure comparing CSI across thresholds would help.**

From the results of the spatial analysis, we confirmed that the CHRain dataset outperformed the ANUClimate and AGCD datasets compared at the daily timestep because it can reproduce more details about rainfall variation and capture high rainfall values better. The bias value of the CHRain dataset compared with the ANUClimate dataset is 0.916.

We added the information in line 487 in the revised manuscript as:

"The spatial evaluation indicated that the CHRain outperforms the ANUClimate and AGCD datasets, which are the most commonly used reliable rainfall datasets in Australia, in representing the 5 km sub-grid rainfall distribution at the Richmond River catchment. The 24-hour total CHRain dataset can also capture high rainfall values better than the ANUClimate (Bias = 0.916)."

**Minor comments**
**Abstract wording: Consider changing "correlation coefficient of 0.949 shows that…" to "During the 2017 event, CHRain achieved r=0.949 against hourly gauges," to anchor the statistic to its context. Daily-alignment note: You already aggregate from 9:00 am–8:00 am to match BoM daily totals; make this alignment explicit the first time daily metrics are mentioned, not just in Methods.**

We adapted all the minor comments in the revised manuscript.

Line 13: "During the 2017 event, CHRain achieved the correlation coefficient of 0.949 against hourly gauges, showing that the dataset can adequately reproduce the patterns of hourly rainfall measurements."

Line 91: "In the areas with sparse distribution of hourly rainfall stations, the daily measurements are disaggregated to hourly data from 9:00 am the previous day to 8:00 am the current day, using patterns from nearby hourly rainfall stations, to match with the daily data provided by the Australian Bureau of Meteorology (BoM)."